# Stable single platinum atoms trapped in sub-nanometer cavities in 12CaO·7Al$_2$O$_3$ for chemoselective hydrogenation of nitroarenes

Tian-Nan Ye [1,6]✉, Zewen Xiao [2,6], Jiang Li [1], Yutong Gong [3], Hitoshi Abe[4,5], Yasuhiro Niwa[4], Masato Sasase[1], Masaaki Kitano [1] & Hideo Hosono[1]✉

Single-atom catalysts (SACs) have attracted significant attention because they exhibit unique catalytic performance due to their ideal structure. However, maintaining atomically dispersed metal under high temperature, while achieving high catalytic activity remains a formidable challenge. In this work, we stabilize single platinum atoms within sub-nanometer surface cavities in well-defined 12CaO·7Al$_2$O$_3$ (C12A7) crystals through theoretical prediction and experimental process. This approach utilizes the interaction of isolated metal anions with the positively charged surface cavities of C12A7, which allows for severe reduction conditions up to 600 °C. The resulting catalyst is stable and highly active toward the selective hydrogenation of nitroarenes with a much higher turnover frequency (up to 25772 h$^{-1}$) than well-studied Pt-based catalysts. The high activity and selectivity result from the formation of stable trapped single Pt atoms, which leads to heterolytic cleavage of hydrogen molecules in a reaction that involves the nitro group being selectively adsorbed on C12A7 surface.

[1] Materials Research Center for Element Strategy, Tokyo Institute of Technology, 4259 Nagatsuta, Midori-ku, Yokohama 226-8503, Japan. [2] Wuhan National Laboratory for Optoelectronics, Huazhong University of Science and Technology, 430074 Wuhan, China. [3] International Center for Materials Discovery, School of Materials Science and Engineering, Northwestern Polytechnical University, 710072 Xi'an, China. [4] High Energy Accelerator Research Organization, KEK, 1-1, Oho, Tsukuba, Ibaraki 305—0801, Japan. [5] Department of Materials Structure Science, School of High Energy Accelerator Science, SOKENDAI, The Graduate University for Advanced Studies, 1-1 Oho, Tsukuba, Ibaraki 305-0801, Japan. [6] These authors contributed equally: Tian-Nan Ye, Zewen Xiao. ✉email: ytn2015@mces.titech.ac.jp; hosono@mces.titech.ac.jp

Supported metal nanostructures have emerged as a well-accepted platform for the development of highly efficient heterogeneous catalysts[1–3]. The particle size of the supported metal plays a critical role in determining the performance of such catalysts[4,5]. Small particles generally provide more active sites and low-coordinated environments that are responsible for excellent catalytic activity[6]. Atomically dispersed metals represent the lower size limit, and maximize atom utilization while reducing the consumption of precious metals[7–10]. However, the controlled synthesis of such single-atom catalysts (SACs) is not easily achieved, because single metal atoms are often unstable and tend to aggregate to form clusters or nanoparticles (NPs) that reduce the large surface energy[11]. Therefore, it is a significant challenge to develop SACs with high thermodynamic stability.

The choice of support is vital for the successful synthesis of SACs, which determines both their thermal stability and durability. Several techniques and different supports have been reported for the preparation of SACs[12–16]. For example, Zhang and colleagues prepared $Pt/FeO_x$ SACs via the substitution of surface Fe atoms by Pt in the crystal lattice of $FeO_x$[7,17]. Yan and colleagues developed a series of polyoxometalate-supported Pt SACs, which gave a quantitative correlation between metal-support interaction and catalytic performance[18–20]. Very recently, ceria has been frequently applied for the preparation of SACs because unsaturated coordinated sites are easily created on its surface, which can act as anchoring sites for single atoms[21,22]. For instance, Fabris and colleagues reported that monodispersed $Pt^{2+}$ ions can be stabilized at monoatomic step edges, which represent the most ubiquitous defects on the surface of reduced $CeO_2(111)$[23]. Recently, López and colleagues firstly discovered the co-existed several charge states of loaded Pt single atoms on $CeO_2$ (100) surface through DFT calculation and first-principle molecular dynamics, which is closely related to the catalytic activity of SACs[24]. For carbon-based supports, Pérez-Ramírez et al. succeeded in anchoring of Pd atoms into the cavities of mesoporous polymeric graphitic carbon nitride, which gave outstanding catalytic activity towards hydrogenation and Suzuki coupling reactions[25–28]. For zeolite systems, Corma et al. demonstrated that single Pt atoms and clusters could be trapped in purely siliceous MCM-22, due to its cup and cage nanostructure. The Pt/MCM-22 catalyst exhibits well-behaved size-selective catalytic activity for the hydrogenation of alkenes[29]. However, suitable supports for the preparation of SACs are still scarce. The surface defects of metal oxide supports are not always sufficiently thermally stable and the anchored single atom species still suffer from sintering issues at high operation temperatures. The single atoms that reside in the inner crystal lattice or the inner porosity of supports always fail to act, which leads to a low atom efficiency. Therefore, a well-structured support material that is rich in exposed stable anchoring sites is highly desirable for the preparation and the long-term operation of SACs.

Nanoporous crystal $12CaO \cdot 7Al_2O_3$ (C12A7) is a compound in the $CaO$-$Al_2O_3$ system that is widely used as the main constituent of aluminous cement[30,31]. However, C12A7 exhibits completely different physical and chemical properties from other compounds in the $CaO$-$Al_2O_3$ system[32]. Most interestingly, it has unique interconnected cage structures with inner diameters of ca. 0.4 nm, which is just the right size for single metal atoms[33–35]. Each unit cage bears a positive charge of +1/3, which is compensated by $O^{2-}$ counter anions that are statistically accommodated within the cages[36,37]. Our recent studies on large surface area C12A7 samples identified its special capability for hydrogen storage under mild conditions due to the abundance of surface sub-nanometer cavities[38]. Isolated metal anion species are thus also expected to be anchored into these surface cavities due to their ideal size and positive charge.

Herein, we report the prediction and synthesis of a stable SAC, Pt/C12A7. The stabilization of Pt single-atoms is achieved by the unique confinement effect of the cavity nanostructure in C12A7, which is confirmed by theoretical simulation and experimental characterizations. The atomically dispersed Pt on C12A7 is demonstrated to have superior stability, even at elevated reduction temperatures up to 600 °C. Pt/C12A7 has excellent catalytic activity and selectivity for the chemoselective hydrogenation of nitroarenes. Multi-technique characterization suggests that the superior catalytic activity and selectivity are due to a cooperative effect between the highly dispersed single Pt atoms for the heterolytic cleavage of $H_2$ and unique surface cavities of C12A7 for the highly preferential adsorption of nitroarenes.

## Results

**The adsorption behaviors of Pt single atom on C12A7 support.** DFT calculations are performed firstly to investigate the energetics of the adsorption models of Pt single atoms on C12A7. Four typical adsorption sites nearby a cavity structure of C12A7 (001) surface and the related adsorption energies ($E_{ads}$) of these anchored Pt single atoms are investigated (Fig. 1a, b). According to the calculated energy profiles, Pt single atom in position 3 is estimated to be the most stable site ($E_{ads} = -9.53$ eV) (Fig. 1b). It should be noted that this $E_{ads}$ is much higher than the cohesive energy of bulk Pt ($-5.85$ eV)[18,22], demonstrating the anchored Pt single atom is thermodynamically stable against sintering, i.e., the aggregation and formation of Pt NPs. Moreover, the formation of Pt dimers and trimers can also be excluded due to the high dimerization and trimerization energy of 2.15 eV $Pt^{-1}$ and 2.61 eV $Pt^{-1}$, respectively (Supplementary Fig. 1)[16,27]. Therefore, we expect the anchored Pt single atom enable to withstand high-temperature conditions. In configuration 3, the Pt single atom was trapped in the middle of a cavity by two exposed oxygen ions, which is quite different from other positions such as configuration 1, coordinated with two oxygen ions outside the cavity; configuration 2, restricted as the cavity wall by two oxygen ions; configuration 4, coordinated with one oxygen ions outside the cavity (Supplementary Fig. 2). Obviously, the surface cavity with the unsaturated oxygen ions of C12A7 provides a unique structure to confine the Pt single atoms firmly (see top and side views in Fig. 1c, d). Therefore, the outlined theoretical modeling predicts that it should be possible to fabricate thermal stable, atomically dispersed Pt on the surface of C12A7.

**Preparation and characterization of the catalyst.** The strategy to prepare the single atom Pt loaded C12A7 catalyst is illustrated in Fig. 1e. The Pt/C12A7 catalyst was fabricated by simple wet impregnation using $K_2PtCl_4$ as precursor. C12A7 has a positively charged lattice framework, $[Ca_{24}Al_{28}O_{64}]^{4+}$, in which the positive charge of the cage wall is compensated for by anions within sub-nanometer cages (0.6 nm diameter, 0.4 nm inner diameter). High surface area C12A7 prepared by the hydrothermal method has a large number of positively charged empty cavities on the surface due to its low crystallinity, as identified by $CO_2$-temperature programmed desorption (TPD) in the below section. The ions in the truncated cages on the top surface can be easily removed by heat treatment under vacuum, generating positively charged cavities with open mouths. This surface property is completely different from that of the bulk, where ions in the cages are surrounded by a Ca-O-Al framework. Therefore, the replacement of the interior ions occurs only under high-temperature conditions (>1000 °C). As a result, $[PtCl_4]^{2-}$ anions (0.47 nm diameter) can be stabilized in the surface cavities by coulomb interaction (Fig. 1e). The atomic Pt would then be directly anchored to the surface cavities of C12A7 upon reduction, which is predicted by

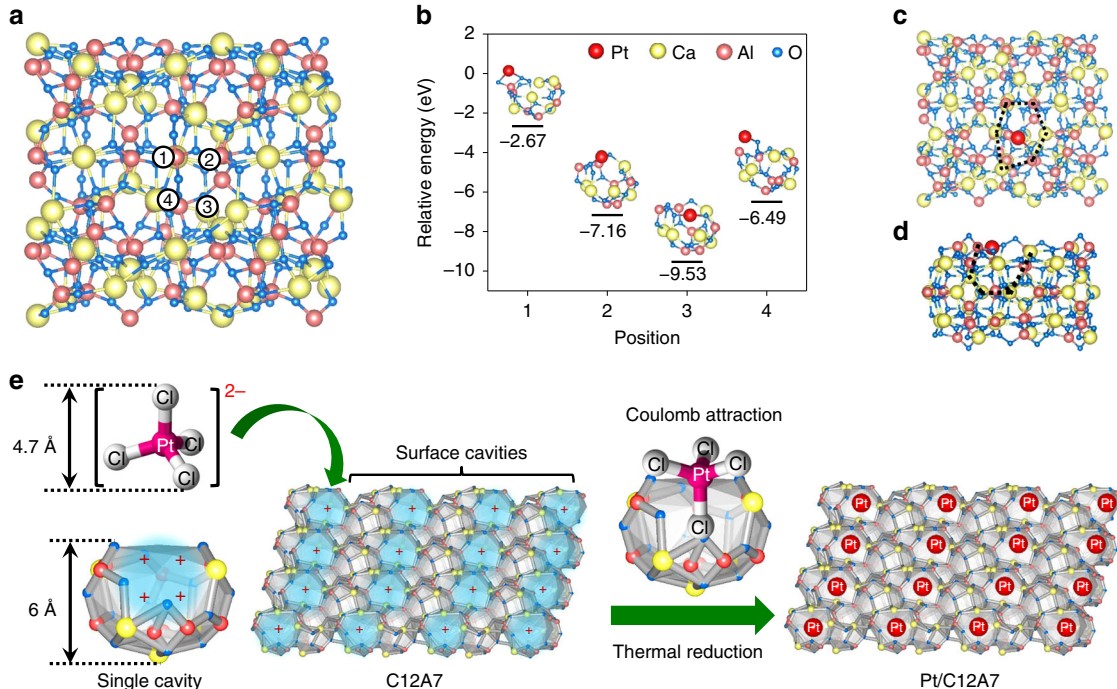

**Fig. 1 Calculated adsorption models and schematic illustration of Pt/C12A7 synthesis. a** Adsorption sites and **b** energetics of the adsorbed Pt single atoms on the (001) surface of C12A7 obtained from DFT calculation. The circles 1–4 in **a** indicate four energy favored adsorption sites nearby a typical cavity structure. Configurations of the most stable anchored Pt single atom on the surface by top (**c**) and side (**d**) view, respectively. Black dotted lines in **c**, **d** indicate open mouths of surface cavities. **e** Interaction between negatively charged $[PtCl_4]^{2-}$ ions and positively charged surface cavities of C12A7 for effective stabilization of single Pt atoms.

the DFT results (Fig. 1). X-ray photoelectron spectroscopy (XPS) measurement confirmed the existence of $Cl^-$ ions (Supplementary Fig. 3), indicating that $Cl^-$ ions replaced the $OH^-$ ions in the sub-cages of C12A7 and stored in them during the reduction process[39].

Aberration-corrected high-angle annular dark-field (HAADF) scanning transmission electron microscopy (STEM) is used to study the morphology of the catalysts. For the 0.1Pt/C12A7 catalyst (Fig. 2a, the preparation details of the various catalysts are shown in Supplementary Table 1), the heavier single Pt atoms could be discerned in the C12A7 support because of the different Z-contrast between Pt and Ca/Al, suggesting that C12A7 stabilizes the Pt species in the form of isolated single atoms, although a small amount of sub-nanometer clusters were also detected. This may be due to the electron beam-induced aggregation of atomic Pt[40]. To further verify that 0.1Pt/C12A7 contained only atomically dispersed Pt throughout the entire catalyst, extended X-ray absorption fine structure (EXAFS) spectra of 0.1Pt/C12A7 shows only Pt-O bonds in the region of 1–2 Å (Fig. 2b), while no Pt-Pt bonds are observed, confirming the sole presence of atomically dispersed Pt on C12A7. The normalized X-ray absorption near-edge structure (XANES) spectra in Supplementary Fig. 4 shows that the peak above the edge (white-line) for 0.1Pt/C12A7 is located between those for Pt foil and $PtO_2$, implying a slight positively charged $Pt^{\delta+}$ rather than $Pt^0$, which agrees with the Bader charge analysis of +0.15 of adsorbed Pt single atom on C12A7 (001). In addition, the coordination number (CN) for Pt-O was estimated to be around 3, which is much smaller than that for coordinatively saturated $PtO_2$ (CN = 6), demonstrating the absence of $PtO_2$ phase in the reduced 0.1Pt/C12A7 (Table 1). As a result, the Pt-O coordination most probably originated from interaction between Pt and unsaturated O of the surface cavities (Fig. 2c, inset), which is consistent with the predicted configurations by DFT calculation

(Fig. 1), suggesting strong metal–support interaction between Pt and C12A7 support. Control catalysts with different supports including $Al_2O_3$ (85 $m^2\,g^{-1}$) and CaO (77 $m^2\,g^{-1}$) are prepared under the same conditions. Only Pt NPs are observed on $Al_2O_3$ and CaO with notable Pt-Pt contributions (Fig. 2b, Table 1 and Supplementary Figs. 5–7).

Inspired by this strategy, we wondered whether the trap effect of the unique surface cavities of C12A7 also applicable to other noble metals. As shown in Supplementary Fig. 8, isolated single atoms such as Ru and Rh dispersed on the C12A7 surface are clearly visible, which can be further confirmed by corresponding EDX and EELs analysis (Supplementary Figs. 9 and 10). These results demonstrated that the single atom trap effect of C12A7 is general toward various transition metals.

The strong electronic interaction between the Pt precursor and the surface cavities in C12A7 plays a pivotal role in the synthesis of SACs. Here we also used $Pt(acac)_2$ as precursor, whose interaction with C12A7 is much weaker than that of $[PtCl_4]^{2-}$ because Pt is stabilized by the acetylacetonate ligands. Figure 2c shows that not only Pt-O but also Pt-Pt metallic bonds appear in $Pt(acac)_2$ derived sample, indicating the formation of Pt clusters or NPs, as confirmed by HAADF-STEM observations (Supplementary Figs. 11, 12). Here, the CN of Pt-O is 1.43 (Table 1), which is much smaller than that prepared from $[PtCl_4]^{2-}$ (CN ~ 3). The fitted distance for the Pt-Pt coordination was estimated to be 2.67 Å, smaller than 2.76 Å of Pt foil, which should be ascribed to contraction of the metal-metal bond when the metal particles are very small[41].

It has been reported that single metal atoms are unstable above 250 °C[29,42]. Interestingly, the single atom species still remained on C12A7 even under a reduction temperature of 600 °C (Supplementary Fig. 13 and Fig. 2d). Here, Pt-O with CN~3.34 of 0.1Pt/C12A7 and CN~2.52 of 0.1Pt/C12A7-R600 can be considered as almost the same coordination environment by

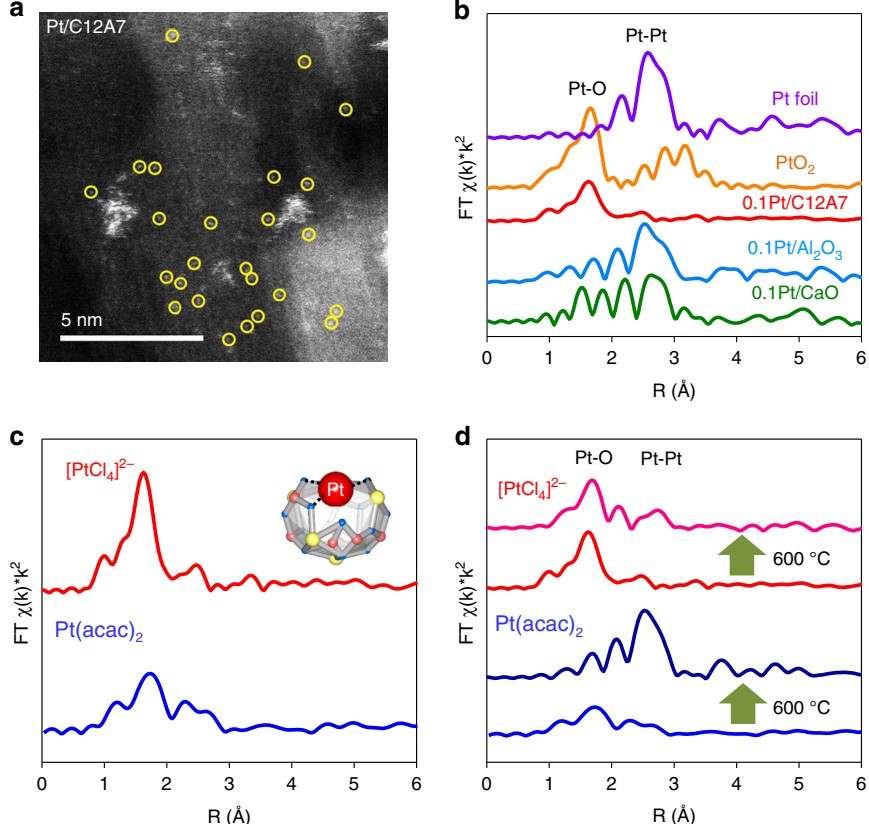

**Fig. 2 Characterizations of Pt/C12A7 with other reference Pt catalysts. a** HAADF-STEM image of 0.1Pt/C12A7 single-atom structures. Single Pt atoms marked in yellow circles are uniformly dispersed on the C12A7 support. **b** Pt K-edge EXAFS spectra in R space for 0.1Pt loaded on various supports, such as Al₂O₃ and CaO. **c**, **d** Pt K-edge EXAFS spectra in R space for 0.1Pt/C12A7 with various Pt precursors and elevated reduction temperatures.

**Table 1 EXAFS fitting results for different Pt catalysts.**

| Sample | Shell | $N$ | $R$ (Å) | $\Delta\sigma^2 \times 10^3$ (Å²) | $\Delta E_0$ (eV) |
|---|---|---|---|---|---|
| 0.1Pt/C12A7 | Pt-O | 3.34 | 2.01 | 2.06 | 14.01 |
| 0.1Pt/C12A7-R600 | Pt-O | 2.52 | 2.03 | 1.43 | 15.73 |
| 0.1acacPt/C12A7 | Pt-O | 1.43 | 2.05 | 0.91 | 14.49 |
|  | Pt-Pt | 1.80 | 2.67 | 5.21 | 3.01 |
| 0.1acacPt/C12A7-R600 | Pt-Pt | 8.64 | 2.74 | 5.61 | 4.67 |
| 0.1Pt/CaO | Pt-Pt | 6.3 | 2.79 | 5.43 | 9.88 |
| 0.1Pt/CaO-R600 | Pt-Pt | 8.02 | 2.76 | 6.03 | 7.56 |
| 0.1Pt/Al₂O₃ | Pt-Pt | 8.69 | 2.75 | 4.81 | 1.84 |
| 0.1Pt/Al₂O₃-R600 | Pt-Pt | 8.57 | 2.77 | 5.47 | 7.21 |

*N coordination number, R distance between absorber and backscattered atoms, $\Delta\sigma2$ disorder term, $\Delta E0$ inner potential correction.*
*Error bounds (accuracies) that characterize the structural parameters obtained by EXAFS spectroscopy are estimated to be N, ±15%; R, ±0.02 Å; $\Delta\sigma^2$, ±20%; $\Delta E^0$, ±20%.*

taking into account the ca. 20% fitting error in EXAFS analysis[19]. In contrast, obvious aggregation and sintering are evident on 0.1acacPt/C12A7-R600 (Supplementary Figs. 14 and 15). In the case of Al₂O₃ and CaO supports without cavity structures, Pt NPs also appeared under high reduction temperatures (Supplementary Figs. 16–18). These results demonstrate that single-atom Pt anchored in the surface cavities of C12A7 support have significant thermal stability under high-temperature reduction conditions. Fourier-transform infrared (FTIR) spectroscopy of CO adsorption behavior are also studied to verify the stable Pt atoms on C12A7. The only adsorption peak with band at 2070 cm⁻¹ of linearly adsorbed CO on $Pt^{\delta+}$ are detected on both 0.1Pt/C12A7 and 0.1Pt/C12A7-R600 (Supplementary Fig. 19). While the signals of CO adsorption on $Pt^0$ sites and bridge sites of Pt clusters or NPs can be observed on 0.1acacPt/C12A7, 0.1Pt/

Al₂O₃, and 0.1Pt/CaO, respectively (Supplementary Fig. 19), but not on 0.1Pt/C12A7 and 0.1Pt/C12A7-R600, demonstrating the atomic dispersion of $Pt^{\delta+}$ species on the surface of C12A7 with exceptional high thermal stability. Here, the CO stretching frequency of 2070 cm⁻¹ on 0.1Pt/C12A7 gave a slightly blue shift compared with other reported SACs, which can be ascribed to the basic surface of C12A7 with a strong electron donation effect, leading to a relative small δ+ value in Pt[43].

**Identified location of atomically dispersed Pt.** To identify the location of the single atom species, CO₂-TPD was conducted to investigate the surface properties of C12A7. For pure C12A7, desorption of CO₂ appeared at 220, 265, and 360 °C in the low temperature region (150–450 °C, Fig. 3a and Supplementary Fig. 20), which can be assigned to different basic sites associated

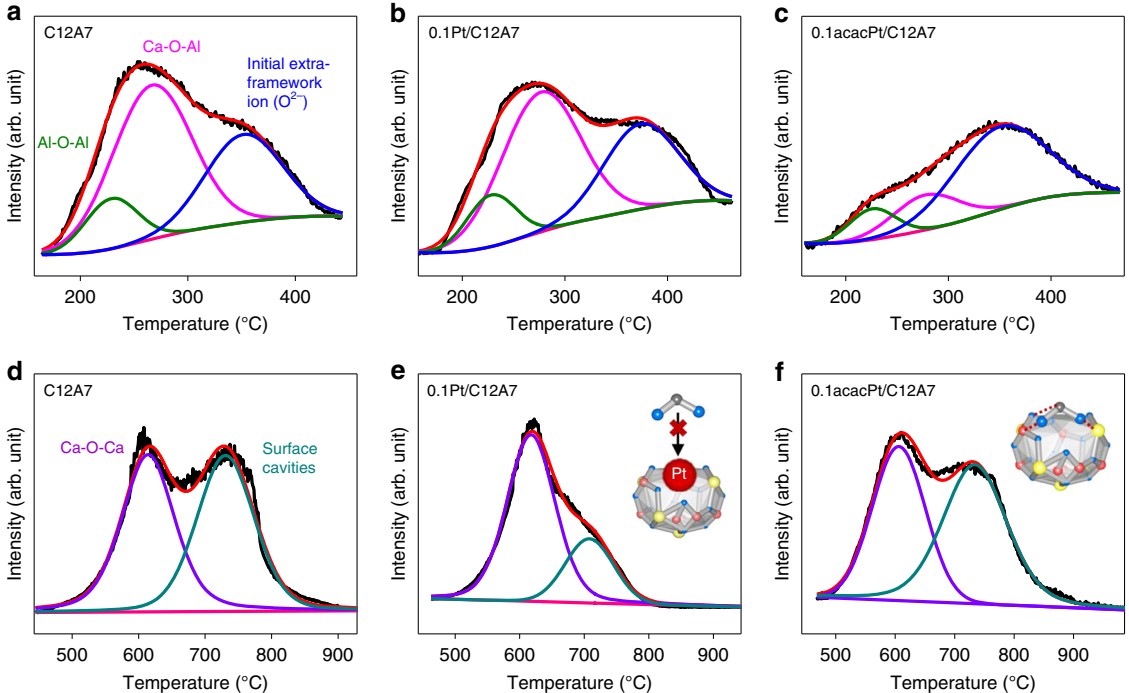

**Fig. 3 CO$_2$-TPD studies of Pt/C12A7 with different Pt precursors.** CO$_2$-TPD profiles for pure **a**, **d** C12A7, **b**, **e** 0.1Pt/C12A7, and **c**, **f** 0.1acacPt/C12A7 in the low and high-temperature regions, respectively. The fitted data for the different temperature regions represent adsorbed CO$_2$ desorption from various kinds of basic sites on the sample surfaces, such as Al-O-Al, Ca-O-Al, initial extra-framework ions (O$^{2-}$), Ca-O-Ca and surface cavities. The difference between the inset of **e**, **f** indicate the suppressed CO$_2$ adsorption in 0.1Pt/C12A7 due to the occupied surface cavities by Pt single atoms.

with the oxygen sites of Al-O-Al, Ca-O-Al and initial extra-framework ions (O$^{2-}$) that merged with the cavity walls, respectively[44]. In the high-temperature region (450–950 °C, Fig. 3d), the basicity mainly originates from two types of oxygen sites, including Ca-O-Ca (615 °C) and surface cavities (725 °C). The Ca-O-Ca sites are ascribed to the small amount of CaO units in the parent C12A7 that cannot be detected in the X-ray diffraction (XRD) patterns (Supplementary Fig. 21), similar to the desorption of CO$_2$ from CaO (Supplementary Fig. 22). The origin of the last desorption peak at 725 °C is considered to be tridentate-adsorbed CO$_2$ on the surface cavities, which indicates strong basicity due to the high configuration energy that was reported in our previous work[45].

The surface properties of C12A7 are generally sensitive to the pyrolysis temperature during the preparation process, and single-phase C12A7 begins to be generated above 500 °C[46]. Thus, three different pyrolysis temperatures (600, 800, and 1000 °C) are studied. The specific surface area of C12A7 gradually reduces from ca. 54 to 8.6 m$^2$ g$^{-1}$ with increasing pyrolysis temperature from 600 to 1000 °C (Supplementary Fig. 23). The narrower and more intense XRD diffraction peaks indicate an increased crystallinity with the elevated pyrolysis temperatures (Supplementary Fig. 21). The absent CO$_2$ desorption peaks of 360 and 725 °C in C12A7-800 and C12A7-1000 demonstrate that the high pyrolysis temperature increases the surface crystallinity and thus forms closed surface cavities, i.e., a reconstructed surface structure (Supplementary Fig. 20)[47]. The surface CaO units are also reduced at high pyrolysis temperatures and almost disappeared at 1000 °C (Supplementary Fig. 20). After loaded with 0.1Pt, C12A7-800, and C12A7-1000 gave much less Pt-O coordination and rich Pt-Pt metallic bonding because of the low surface areas and lack of surface cavities (Supplementary Fig. 24).

The location of the single Pt atoms could be accurately identified by CO$_2$-TPD (Fig. 3 and Supplementary Fig. 25). As indicated in Fig. 3a, b, there is almost no difference between C12A7 and 0.1Pt/C12A7 in the low temperature region (150–450 °C). However, tridentate adsorption of CO$_2$ on the surface cavities (725 °C) is suppressed in 0.1Pt/C12A7 (Fig. 3e), which should be ascribed to the domination of Pt single atom (Fig. 3e inset). In contrast, different CO$_2$ adsorption behavior of 0.1acacPt/C12A7 appear in the low temperature region with almost no change in the high-temperature region compared to pure C12A7 (Fig. 3c, f), indicating the accumulation of Pt clusters or NPs on surface Al-O-Al and Ca-O-Al sites of C12A7 instead of the surface cavities. Therefore, single Pt atoms should be confined in the unique surface cavities of C12A7. Here, the amount of adsorbed CO$_2$ molecules can be estimated through the calibration of CO$_2$ standard gas by mass spectrometry. According to our previous work of CO$_2$ adsorption on C12A7:e$^-$, each surface truncated cage only corresponds to one CO$_2$ molecule (Fig. 3f inset), which can be used to determine the concentration of surface truncated cages of C12A7. As shown in Supplementary Table 2, the amount of surface truncated cage is ca. 9.1 μmol g$^{-1}$, which means the upper limit of the single atomic Pt is 1.8 mg$_{Pt}$ g$^{-1}$$_{catalyst}$ (~0.18 wt%).

**Catalytic performance of the catalysts.** The catalytic performance test start from the hydrogenation of 4-chloronitrobenzene because the selective product of 4-chloroaniline is of critical importance in catalyzed cross-coupling reactions[48–51]. As shown in Supplementary Fig. 26, 0.1Pt/C12A7 affords the exclusive formation of 4-chloroaniline with high activity (99.9%, conversion) in 2 h under 0.5 MPa H$_2$ at 60 °C. Even at room temperature, the hydrogenation of 4-chloronitrobenzene proceeds smoothly with slight longer reaction time (9 h, Supplementary Table 3). And the turnover frequency (TOFs) of 0.1Pt/C12A7 is estimated to be 25772 h$^{-1}$, which is one order of magnitude higher than those of 0.1Pt/CaO (3396 h$^{-1}$) and 0.1Pt/Al$_2$O$_3$ (1178 h$^{-1}$) under the same reaction conditions (Fig. 4a). Notably,

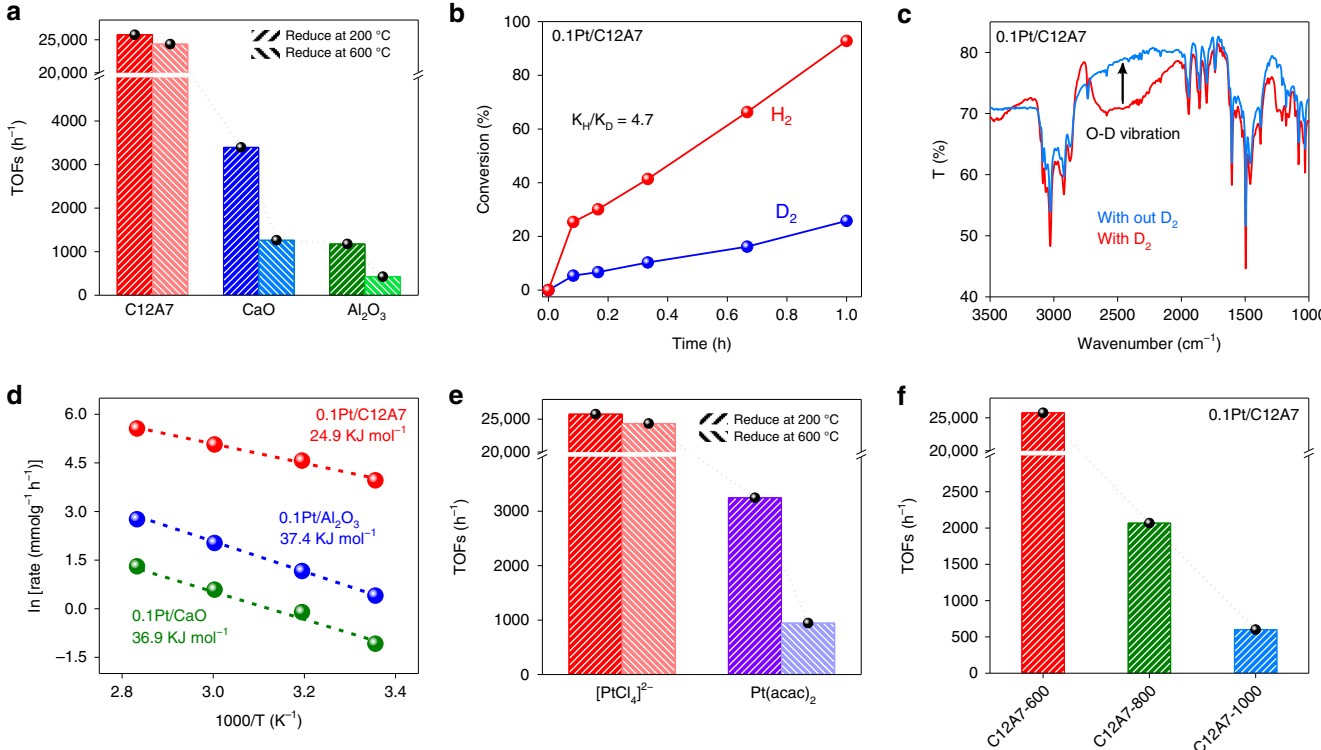

**Fig. 4 Catalytic performance and related kinetic studies. a** Plot of 0.1Pt loaded on various supports at elevated reduction temperatures and their TOFs for the hydrogenation of 4-chloronitrobenzene. **b** Primary isotope effect observed on 0.1Pt/C12A7 for the hydrogenation of 4-chloronitrobenzene. **c** FTIR spectra for 0.1Pt/C12A7 after exposure to $D_2$ with the arrow pointing the O-D vibration. This vibration disappears after the introduction of styrene. **d** Arrhenius plots for 0.1Pt loaded on various supports. **e** Plot of 0.1Pt/C12A7 with different Pt precursors at elevated reduction temperatures and their TOFs for the hydrogenation of 4-chloronitrobenzene. **f** Plot of 0.1Pt loaded on C12A7 support prepared at different calcination temperatures and their TOFs for the hydrogenation of 4-chloronitrobenzene. The TOFs was calculated from the reaction rate at a low conversion level. Reaction conditions: 5 mg catalyst, 0.5 mmol 4-chloronitrobenzene, 5 ml methanol, 60 °C, 0.5 MPa $H_2$. The yields given below the structure were determined using GC and GC-MS with n-hexadecane as an internal standard.

the TOFs of 0.1Pt/C12A7-R600 remained largely unchanged ($24351\ h^{-1}$) compared with 0.1Pt/C12A7, which is quite different from 0.1Pt/CaO and 0.1Pt/Al$_2$O$_3$ catalysts with deactivation of ca. 60% when the reduction temperature raised to 600 °C (Fig. 4a). Here, 0.1Pt/C12A7 also offers high activity towards various chloronitrobenzenes hydrogenation reactions (Supplementary Tables 4–6), surpassing other reported Pt-based heterogeneous catalysts under similar reaction conditions.

Mass loading effect is also studied and the selectivity for 4-chloroaniline decreased with increasing Pt amount (Supplementary Table 3). In these cases, the unavoidable concomitant formation of dehalogenation products readily occurs, which should be attributed to the disparate adsorption behavior of the substrates on large Pt particles (Supplementary Figs. 27–29). For single Pt atoms, 4-chloronitrobenzene may preferentially anchor perpendicularly at the top of the C12A7 support or Pt species only through oxygen atoms instead of aromatic rings. In contrast, the adsorption of 4-chloronitrobenzene on large Pt particles occurred preferentially via the aromatic ring rather than the nitro group[52]. These results demonstrat that isolated Pt atoms are distinguishable from the surface of C12A7 until the Pt loading decreases to 0.12 wt%.

To further confirm the catalytic performance of 0.1Pt/C12A7 at high temperature, CO oxidation reaction is also investigated. As shown in Supplementary Fig. 30a, 0.1Pt/C12A7 and 0.1Pt/C12A7-R600 exhibit similar trends in terms of conversion and reached 100% CO conversion at around 225 °C. Moreover, the catalytic stability of 0.1Pt/C12A7 catalyst is also investigated in a

continuous flow of the reactant gas at 175 and 250 °C for 80 h, respectively (Supplementary Fig. 30b). In both stability experiments, no obvious deactivation occurred, illustrating the high-temperature stability of 0.1Pt/C12A7 SACs even under oxidative condition.

**Kinetic analysis and interaction between reactants and catalysts.** The catalytic kinetic isotope effect (KIE) and Fourier transform infrared (FTIR) spectroscopy are powerful techniques to elucidate the manner of $H_2$ dissociation and thus reveal the reaction mechanism. Using $D_2$ for 4-chloronitrobenzene hydrogenation, the reaction rates of 0.1Pt/CaO and 0.1Pt/Al$_2$O$_3$ slow down by a factor of 1.5–1.6 (Supplementary Fig. 31) due to the zero-point energy difference between the isotopic isomers[8]. However, the primary isotope effect is as large as 4.7 ($k_H/k_D$) over 0.1Pt/C12A7 (Fig. 4b), indicating the change of reaction rate-controlling step (RDS). In FTIR spectra, O-D stretching vibration can be detected on 0.1Pt/C12A7 but absent on 0.1Pt/CaO and 0.1Pt/Al$_2$O$_3$ (Fig. 4c and Supplementary Figs. 32 and 33). Here, heterolytic cleavage of $H_2$ tend to generate $H^{\delta+}$ and $H^{\delta-}$ at the Pt-O interface of 0.1Pt/C12A7. Thus, the RDS for the hydrogenation reaction over 0.1Pt/C12A7 may involve the cleavage of O-H bonds instead of Pt-H bond breaking[8,53]. Such heterolytic cleavage of $H_2$ would be favored for chemoselective reduction of polar functionalities such as nitro groups[38,54]. As a consequence, 0.1Pt/C12A7 catalyst gives a relative lower activation energy towards the hydrogenation reaction (Fig. 4d). In addition,

Supplementary Fig. 34 shows the dependency of the initial reaction rates on the partial pressures of $H_2$ and 4-chloronitrobenzene concentration. The reaction rates over 0.1Pt/C12A7 are more sensitive to the hydrogen pressure compared with the 4-chloronitrobenzene concentration, which results in kinetic reaction orders of $\alpha(H_2) = 0.94$ and $\beta$(4-chloronitrobenzene) = 0.36, respectively. Compared with 0.1Pt/CaO, 0.1Pt/Al$_2$O$_3$, and 0.1acacPt/C12A7, 0.1Pt/C12A7 catalyst with single Pt atoms provides more efficient active sites for heterolytic hydrogen dissociation, which accelerates the hydrogenation reaction and delivers higher catalytic activity (Fig. 4a, e). Here, the hydrogen activation process may also be promoted by the Pt single atoms with low CN of Pt-O in the surface cavities of C12A7, which is in good agreement with Pt$_1$/Fe$_2$O$_3$[55]. The 0.1Pt/C12A7-800 and 0.1Pt/C12A7-1000 catalysts without single atomic species and with lower surface alkalinity exhibit much poorer catalytic performance, as shown in Fig. 4f.

High catalytic performance of heterogeneous catalysts generally depends on a strong adsorption capability for the reactants with a weak adsorption capability for the products. To understand the interactions between reactants/products and our catalysts, the adsorption behavior of nitrobenzene is evaluated using TPD-diffuse reflectance infrared Fourier transform (DRIFT) spectroscopy. Supplementary Fig. 35 shows bands at 1527 and 1349 cm$^{-1}$, associated with asymmetric stretching ($\nu_{as}$) and symmetric stretching ($\nu_s$) vibration frequencies of nitro groups, respectively[56,57]. Notably, the intensity of the 1349 cm$^{-1}$ bands for CaO and Al$_2$O$_3$ reduce gradually with the increased vacuum and cell temperature. In contrast, the 1349 cm$^{-1}$ bands of C12A7 decrease slowly with an obvious plateau at $10^{-3}$ Pa, indicating a strong interaction of nitrobenzene molecules on the C12A7 surface, in which the two oxygen atoms of the nitro group may interact with the unsaturated Ca or Al atoms of the surface cavities. On the other hand, aniline desorbed much faster from C12A7 than CaO and Al$_2$O$_3$ (Supplementary Fig. 36). Moreover, the partial density of states (PDOS) of Pt atom on surface of C12A7 is also calculated and the Pt atoms with up-shift d-band states are much close to the Fermi level (Supplementary Fig. 37), resulting much stronger adsorption capability for the reactants such as $H_2$ and nitroarenes[58]. Both experimental TPD-DRIFT and calculated d-band center shift are consistent with the superior catalytic performance of 0.1Pt/C12A7.

## Catalytic stability and scope of hydrogenation of nitroarenes.

The stability and reusability of the catalyst were carefully examined by cycle experiments and further characterization. The 0.1Pt/C12A7 catalyst could be reused at least four times (Supplementary Fig. 38) without obvious attenuation of activity. No destruction of the structure was evident from XRD or EXAFS measurements, demonstrating the robust stability of 0.1Pt/C12A7 (Supplementary Figs. 39, 40). Note that the reaction proceeds only in the presence of 0.1Pt/C12A7, and no more products could be confirmed after removal of the catalyst from the reaction process, which rules out any possible contribution of homogeneous catalysis by leached Pt species (Supplementary Fig. 41). Further investigation of the substrate scope was performed under the same reaction conditions. As summarized in Table 2, 0.1Pt/C12A7 also showed high activity for a broad scope of substituted nitroarenes containing −OH, −CF$_3$, −CN, −CO, −COC, −COOC or −CON functional groups.

## Discussion

We anchor single Pt atoms into the unique surface subnanometer cavities of C12A7 with exceptionally high thermal stability. The key factor for the synthesis of the stable single atom is the strong interaction between the Pt anions and the positively charged surface cavities of C12A7 with the just right size. The surface unsaturated oxygen sites of the cavities coordinated with the single Pt atoms, which proved by theoretical predictions and experimental characterizations, significantly enhancing the metal–support interactions. The 0.1Pt/C12A7 catalyzes the hydrogenation of nitroarenes to anilines with excellent activity and selectivity. The superior catalytic performance is attributed to the synergistic effect between the highly dispersed single Pt atoms and the unique surface properties of the C12A7 support. Here, this synthetic methodology would be a versatile method for the production of various stable SACs, which are expected to exhibit excellent catalytic performance for hydrogenation reactions or other catalytic processes.

## Methods

**Synthesis of C12A7**. C12A7 was synthesized by a hydrothermal method followed by calcination. Briefly, a stoichiometric mixture (Ca:Al = 12:14) of Ca(OH)$_2$ and Al(OH)$_3$ was dispersed in distilled water and ball-milled at a speed of $12.1 \times g$ for 4 h. The mixture was then hydrothermally treated in a Teflon-lined stainless autoclave at 150 °C for 6 h with stirring. The product was separated by centrifugation and calcined at 600 °C for 5 h in air. The C12A7 sample was then collected as a white powder. C12A7-800 and C12A7-1000 powders were obtained by calcination at 800 and 1000 °C, respectively.

**Synthesis of Pt/C12A7**. Pt/C12A7 was prepared with a conventional wet impregnation method, in which K$_2$PtCl$_4$ was used as the Pt precursor and methanol served as the solvent. Prior to deposition, the C12A7 powder was pretreated at 500 °C for 5 h in vacuum (~1 × $10^{-4}$ Pa) to remove water and oxygen adsorbed on the surface. The evacuated C12A7 powder was dispersed in a solution of K$_2$PtCl$_4$ in methanol and continuously stirred at room temperature for 12 h under an Ar atmosphere. After evaporation of the solvent at 40 °C under vacuum, the obtained powder was treated in 5% H$_2$/Ar at 200 °C to reduce the Pt species. For other transition metal based SACs of C12A7, (NH$_4$)$_2$RuCl$_6$ and Na$_3$RhCl$_6$ were used as Ru and Rh precursors, which were conducted according to the similar thermal reduction method as that used for Pt.

**Synthesis of Pt/CaO, Pt/Al$_2$O$_3$, acacPt/C12A7**. Pt nanoparticle catalysts supported on CaO, Al$_2$O$_3$, and C12A7 were also prepared by a wet impregnation method. Briefly, the supports were pretreated at 500 °C for 5 h in a vacuum (~1 × $10^{-4}$ Pa) and then dispersed in a methanol solution with an appropriate amount of K$_2$PtCl$_4$ or Pt(acac)$_2$ at room temperature under an Ar atmosphere. After stirring for 12 h, methanol was evaporated at 40 °C under vacuum and the solids were further treated in 5% H$_2$/Ar at 200 °C to reduce the Pt species.

**Catalytic hydrogenation of nitroarenes**. All reactions were conducted in a 25 mL stainless steel autoclave fitted with a glass tube, a 60 bar manometer, and a magnetic stirrer. In a typical reaction run, 0.5 mmol of nitroarenes and 5 mg of catalyst were mixed in 5 mL of solvent. The autoclave was then flushed three times with H$_2$, pressurized with H$_2$ (0.5 MPa), and reactions were performed in the autoclave at an appropriate temperature. The products were analyzed using gas chromatography (GC) and n-hexadecane was used as an internal standard. The composition of the products was further confirmed by GC-mass spectrometry (GC-MS). The TOF was separately measured by keeping the substrate conversion below 20%. The TOF values were calculated in the format of mol$_{4\text{-chloroaniline}}$ mol$^{-1}_{\text{total Pt}}$ h$^{-1}_{\text{reaction time}}$ and the amount of metal is based on the moles of total Pt involved in the catalyst.

**CO oxidation**. CO oxidation experiments were performed in a fixed-bed flow system using a quartz-tube reactor with a flow of 34 mL·min$^{-1}$ 4.72% O$_2$/He and 6.9 mL·min$^{-1}$ 9.52% CO/He (WHSV = 25,200 mL·g$^{-1}$·h$^{-1}$). In the standard procedure, 0.1 g Pt/C12A7 catalyst was set in the reactor with quartz wool and pretreated in a stream of H$_2$ and N$_2$ at 250 °C. After the pre-treatment, the temperature was increased to the operation temperature at the rate of 6 °C min$^{-1}$ and then held for 20 mins before the measurement of the steady-state conversion level. The reactants and products were analyzed by online gas chromatography (490-MicroGC, Agilent) using a thermal conductivity detector.

**Characterization**. Samples were examined by aberration-corrected high-angle annular dark-field (HAADF) scanning transmission electron microscopy (STEM; JEM ARM-200F, JEOL) at 200 kV, after deposition of the sample as a dry powder onto a holey carbon film supported by a 300 copper mesh. The mean Pt particle diameters were calculated by measuring the size of more than 100 particles per sample. X-ray absorption fine structure (XAFS) measurements were performed on the BL-12C beamline (Photon Factory at the Institute of Materials Structure Science, High Energy Accelerator Research Organization, Tsukuba, Japan). A Si(111)

**Table 2 Selective hydrogenation of substituted nitroarenes over the 0.1Pt/C12A7 catalyst.**

| Entry | Substrate | Product | Time (h) | Conv. (%) | Select. (%) |
|---|---|---|---|---|---|
| 1 | Cl — C6H3 — NO2 | Cl — C6H3 — NH2 | 2 | 99.9 | 99.4 |
| 2 | NO2 / Cl (ortho) | NH2 / Cl (ortho) | 2 | 99.9 | 97.6 |
| 3 | HO — C6H4 — NO2 | HO — C6H4 — NH2 | 1.5 | 99.9 | 99.9 |
| 4 | F3C — C6H4 — NO2 | F3C — C6H4 — NH2 | 2 | 99.9 | 99.9 |
| 5 | N≡C — C6H4 — NO2 | N≡C — C6H4 — NH2 | 3 | 99.9 | 96.4 |
| 6 | CH2=CH — C6H4 — NO2 | CH2=CH — C6H4 — NH2 | 2.5 | 99.9 | 93.1 |
| 7 | CH3CO — C6H4 — NO2 | CH3CO — C6H4 — NH2 | 2 | 99.9 | 97.1 |
| 8 | CH3O2C — C6H4 — NO2 | CH3O2C — C6H4 — NH2 | 2 | 99.9 | 99.9 |
| 9 | H2NOC — C6H4 — NO2 | H2NOC — C6H4 — NH2 | 2 | 99.9 | 99.9 |

Reaction conditions: 0.5 mmol substrate, 5 mg catalyst, 5 mL methanol, 60 °C, 0.5 MPa $H_2$. Conversion (Conv.), and selectivity (Select.) were determined by GC and GC-MS using n-hexadecane as an internal standard.

double-crystal monochromator was used to obtain the monochromatized X-ray beam, and spectra were obtained in transmission mode and fluorescence yield mode for diluted samples. The sample was pressed with a hand-press apparatus to obtain a pellet sample. The pellet sample was then sealed in a plastic bag for measurement. XAFS spectra were analyzed using the Athena and Artemis software packages. The FEFF6 code was used to calculate the theoretical spectra. The Pt content was determined using inductively coupled plasma atomic emission spectroscopy (ICP-AES; ICPS-8100, Shimadzu). Nitrogen sorption measurements were performed to evaluate the Brunauer-Emmett-Teller (BET) surface areas of the catalysts using an adsorption analyzer (BELSORP-mini II, BEL, Japan). The crystal structure was analyzed using X-ray diffraction (XRD; D8 Advance, Bruker) with monochromated Cu Kα radiation ($\lambda = 0.15418$ nm).

**Investigation of the surface basic sites of C12A7**. Carbon dioxide temperature programmed desorption (CO2-TPD; BELCAT-A, MiccrotracBEL, Japan) profiles were measured to determine the surface basic sites of C12A7. Samples (50 mg) were placed in a U-shaped quartz reactor with an inner diameter of 0.5 cm, exposed to a flow of $CO_2$ at 300 °C for 30 min, and then cooled to room temperature. TPD was subsequently measured with a heating rate of 10 °C min$^{-1}$ until 1000 °C. The

desorption products were analyzed with a thermal conductivity detector (TCD) and mass spectrometer (Bell Mass, MiccrotracBEL, Japan).

**Investigation of the isotope effect**. The thin-liquid film method with $CaF_2$ windows was used to characterize the formation of −OD bonds. The freshly prepared catalysts were typically dispersed in 10 mL of toluene in a glass vessel and then 0.1 MPa of $D_2$ was introduced into the vessel. The suspension was sequentially stirring for 2 h at room temperature. After the reaction, the suspension was subjected to FTIR spectroscopy examination (Nicolet 6700, Thermo Scientific). For 0.1Pt/C12A7, the −OD bond vibration was evidenced by a broad peak at 2600 cm$^{-1}$. After that, microliters of styrene were introduced into the −OD bearing 0.1Pt/C12A7 and continuously stirred for 2 h at room temperature. After the reaction, the –OD signal disappeared for 0.1Pt/C12A7.

**Adsorption behavior of CO molecules, reactants and products**. Diffuse reflectance infrared Fourier transform spectroscopy (DRIFTS) were performed on a spectrometer (FT/IR-6100, Jasco) with a Hg-Cd-Te (MCT) detector at 4 cm$^{-1}$ resolution. Typically, 30 mg catalyst was put into an alumina sample cup and settled

into a stainless steel heat chamber with cooling water (STJ-0123-HP-LTV, S.T. Japan). Then the chamber was covered by a KBr window. The catalyst was pretreated under vacuum at 200 °C for 2 h before the measurement. After being cooled to room temperature, pure CO (99.99999%) was supplied to the system. In the case of reactants and products, nitrobenzene or aniline were fed instead. The infrared spectrum of the sample at room temperature prior to the adsorption was collected as the background for the difference spectra obtained by subtracting the background from the spectrum of a sample with an adsorbed substrate.

**DFT calculations**. The geometric and electronic structure calculations were performed using the density functional theory (DFT) as implemented in the Vienna Ab initio Simulation Package (VASP). The generalized gradient approximation (GGA) with the Perdew-Burke-Ernzerhof (PBE)[59] functional and the projector augmented plane-wave (PAW) method were employed[60]. 500 eV was set as the plane-wave cutoff energy. Γ-centered k-meshes with a k-spacing of 0.2 Å$^{-1}$ were adopted to sample the Brillouin zones. The force convergence criterion for the structure relaxation was 0.01 eV·Å$^{-1}$. The C12A7 (001) surface was constructed based on the optimized bulk lattice parameters, with a periodic supercell containing a vacuum width of 15 and a 12.1 Å thick slab with a $1 \times 1$ lateral unit cell.

The central one third of atoms of the surface model were kept fixed to hold the characteristics of realistic surface, while the rest atoms were allowed to be fully relaxed during the geometry optimizations. A Bader charge analysis was conducted using the Bader program[61]. The adsorption energies of Pt single atoms, dimers and trimers on the C12A7 (001) surface were calculated using the following equation (Eq. 1):

$$E_{ads} = E_{total}(Pt_n/C12A7) - nE(Pt) - E_{total}(surface) \qquad (1)$$

where $E_{total}(Pt_n/C12A7)$ is total energy of the optimized Pt single atom ($n = 1$), dimer ($n = 2$), or trimer ($n = 3$) on the surface, $E(Pt)$ is the total energy of a Pt single atom, and $E_{total}(surface)$ is the total energy of the surface model.

## Data availability
The data that support the findings of this study are available from the corresponding authors on reasonable request.

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

## Acknowledgements

This work was supported by a MEXT Element Strategy Initiative to Form Core Research Center (Grant Number JPMXP0112101001). A part of this work was supported by a PRESTO Grant (No. JPMJPR18T6) from the Japan Science and Technology Agency (JST) and Kakenhi Grants-in-Aid (Nos. 17H06153, JP19H05051 and JP19H02512) from the Japan Society for the Promotion of Science (JSPS). T-N.Y. is supported by a JSPS fellowship for International Research Fellows (No. P18361). We also thank Prof. Hara and Dr. Hattori (Tokyo Institute of Technology) for the help with XPS measurement.

## Author contributions

H.H. and T-N.Y. proposed the idea behind the research. H.H. supervised the project. T-N.Y., J.L., and Y.G. performed the synthesis, characterization and catalytic measurements. Z.X. carried out the model construction and DFT calculations. H.A. and Y.N. helped with the X-ray absorption fine-structure measurements. M.S. performed the HAADF-STEM measurements. Y.T., Y.G., M.K., and H.H. co-wrote the paper. All authors discussed the results and commented on the paper.

## Competing interests

The authors declare no competing interests.
