## [Peer Review File · Nature Communications]

Reviewers' comments:

Reviewer #1 (Remarks to the Author):

The strength of this work lies in the use of a new support material, C12A7, which has a unique surface sub-nanometer cavity as the anchoring sites for Pt single atoms. The resultant single-atom catalyst has excellent stability against high-temperature reduction, which is not easily achieved with other reported single-atom catalysts. In this respect, I think this work is worth of publication in Nat Comm. However, there are some major concerns to be addressed before consideration of publication.

1. The EXAFS results show the coordination number of Pt-O shell in the 0.1Pt/C12A7 is 3.34, while the DFT calculation indicates that the Pt single atom is trapped in the middle of a cavity by two exposed oxygen ions (line 107). They are inconsistent.

2. Please explain why the activity (TOF value) of the 0.1Pt/C12A7-R600 was almost the same as that of the 0.1Pt/C12A7-R200. This is difficult to understand if you consider that the two catalysts have different Pt coordination number and oxidation state, which, I believe, would affect greatly the hydrogenation activity.

3. Please show how you calculated the TOF value. For example, by using the data in Table S2, the TOF value of the 0.1Pt/C12A7 was only 8125 h⁻¹ (what I calculated), rather than 25772 h⁻¹. Moreover, the comparison with literature (Table S3-S5) should be at similar reaction conditions. For the fair comparison, you should use the reaction data at 2 h rather than the initial several minutes. For this reason, your claim that "surpassing the catalytic performance of all previously reported Pt-based heterogeneous catalysts under similar conditions" (line 293-294) needs to be revised.

4. In table S2, for the samples with increased Pt loadings, the activity might be higher. In this case, the reaction time of 2 h might be so long that the side reactions took place. For the selectivity comparison at similar conversion, the reaction time might be varied depending on the Pt loading.

5. Grammar errors:

Line 69, "is highly desirable for the preparation of and the long-term operation of SACs."

Line 204, "The coordination number around Pt is 1.45 (Table 1)." The description is not accurate because there are two shells in Table 1, Pt-O and Pt-Pt. Please indicate which shell. Moreover, the figure in Table 1 is 1.43, rather than 1.45.

Reviewer #2 (Remarks to the Author):

The paper reports an interesting system where stable Pt single-atoms are trapped in the within sub-nanometer surface cavities in well-defined 12CaO·7Al₂O₃ (C12A7) crystals. The catalyst exhibit high activity and selectivity in nitro-group hydrogenation reactions. It may become publishable in Nat. Commun., but existing issues, as detailed below, defer the reviewer to come up with a final recommendation.

1) Recently, the relationship between the metal-support strength and single Pt atom stability/activity has been systematically studied for the first time (JACS, 2019, 141, 8185-8197.) Two major findings in the paper include 1) the least adsorption energy to obtain sintering resistant Pt SACs is 5.50 eV, exactly matching the cohesive energy of bulk Pt metal; and 2) stable Pt SACs may not necessarily be less active due to several competing factors. The current study is a nice example to illustrate the correctness of the earlier study. First, the adsorption energy of Pt in the cavity is -9 eV, far more than the cohesive energy of bulk Pt. Also, such stable Pt SACs are very active in nitro reduction reaction. Therefore, credit should be given to the JACS paper where quantitative correlation between Pt-support interaction and stability/hydrogenation activity have already been given.

2) The major issue with the scientific part is the lack of evidence for the hydride mechanism. A pronounced KIE effect (>4) only suggest the rate-determining step in the Pt SAC catalyst is different

from the two other samples. It can not be used as a proof for the existence of the hydride mechanism. To the minimum, the authors should conduct further DFT calculations to show that the hydride might be the preferred pathway.

3) There is a lack of reaction kinetic information. The authors select conditions where near 100% conversions are achieved, and therefore hide the intrinsic kinetic data. The reaction order for both nitrocompounds and H₂ should be measured.

4) The CO adsorption on all Pt samples should be provided. This provides additional clues whether Pt species are truly fully atomically dispersed. HRTEM only tells a tiny fraction of the samples while XAS is not sensitive enough to detect a small amount of NPs (for example, less than 10%).

5) What also puzzles the reviewer, is that while the major advantage of the catalyst is the high temperature stability under reducing condition, the catalyst is only used for nitro group reduction reactions at 60 degree C. Why not trying more demanding reactions that require 200 degree C and above, under H₂ atmosphere?

6) Based on Figure 1, the C12A7 material should be able to support single-atom Pt in a very high loading, if each site can indeed stabilize one Pt atom. In reality, a loading of 0.1 wt% was used. How do we understand this? If only stable Pt SAC could only be prepared under such a low loading, the reviewer has reasons to doubt that Pt may not locate on C12A7 as the authors suggested.

7) It would be nice that at least the authors comment whether their support is able to anchor metal atoms beyond Pt.

8) Finally, the single-atom catalysis research is a fast growing area. Some relevant papers, such as Pt SAC for nitro group reduction (e.g., *Angew. Chem. Int. Ed.*, 2016, 55, 8319–8323.) and review articles on enhancing the stability of SACs (e.g., *Adv. Mater.*, 2018, 30, 1802304.) should be cited.

Reviewer #3 (Remarks to the Author):

The authors present the the formation of single atoms on an CaO.AIO system for the hydrogenation of nitrobenzene. The work might be relevant but the results are sparse. Complementary data is needed, fundamentally on the computational side to show the robustness of the present approach. Therefore, I cannot recommend the manuscript for publication in its present form.

The list of classes of SAC is too short. Isolated atoms on other metals and on carbons are by far one of the most investigated.

The use of mixed units (eV and kJ/mol) should be avoided.

Energies of -10 eV like Pt adsorption basically means that the zero reference is completely misplaced. A better reference needs to be chosen. Particularly, when Figure 1 reports PtCl₂- as reference.

The speciation has shown to be crucial for the reactivity (see *ACIE* and *Nature Comm.* Perez-Ramirez 2019). Dimerization energies need to be calculated much before aggregation can be discarded only on terms of energies. Pt-Pt distances can be larger than 2-3 Angstrom found in the experiments (see discussion *ACIE* 2019).

The description of the atoms needs some electronic density analysis. Bader charges, DOS, XPS shifts all of these characterize properly the structure of the atoms.

How do the authors consider the elimination of Cl from the precursor?

The hydrogenation of nitrobenzene has been reported in the literature supported by detailed

calculations (Perez-Ramirez, ChemCatChem, ACS Catal). The authors can map their discussion to the previous description in the literature. The full reaction can be calculated to illustrate the enhancement.

What is the reaction order of H₂? and of nitrobenzene? This can help in the investigation.

Charges on atoms can change depending on the oxide (Daelman Nature Mat. 2019) are the authors sure that this does not contribute to their work.

Response Letter

Reviewer #1 (Remarks to the Author):

1. The EXAFS results show the coordination number of Pt-O shell in the 0.1Pt/C12A7 is 3.34, while the DFT calculation indicates that the Pt single atom is trapped in the middle of a cavity by two exposed oxygen ions (line 107). They are inconsistent.

R: We appreciate the referee for attracting our attention to this issue. Normally, there is ca. 20% fitting error in EXAFS analysis which is also pointed out by the literature.¹ It leads to a bit difference between the DFT calculation results and the EXAFS fitting values, which is common in other reported single atom catalysts (SACs) systems such as Pt-PMA/AC and Fe-ISAs/CN.^{1,2}

Actually, the coordination number fitted from the EXAFS results was an average number of all probable Pt-O coordination states. In our case, the EXAFS results derived Pt-O coordination number was estimated to be 3.34 and 2.25 in 0.1Pt/C12A7 and 0.1Pt/C12A7-R600 respectively, indicating the real coordination number of Pt-O should be 2 or 3. While DFT calculation points out that the Pt single atom is trapped by two exposed oxygen ions, which is still consistent with the fitted coordination number from EXAFS results. Most importantly, DFT calculations predict that Pt atom is solely trapped in the middle of a cavity which is highly consistent with the experimental CO₂-TPD results of the confined Pt single atoms in unique surface cavities of C12A7 (Fig. 3).

2. Please explain why the activity (TOF value) of the 0.1Pt/C12A7-R600 was almost the same as that of the 0.1Pt/C12A7-R200. This is difficult to understand if you consider that the two catalysts have different Pt coordination number and oxidation state, which, I believe, would affect greatly the hydrogenation activity.

R: We appreciate the referee for attracting our attention to this issue. Indeed, the Pt-O coordination number directly affects the hydrogenation activity as well as the selectivity over Pt

SACs, which was reported in a recent published work.³ In our work, Pt single atom with relative low coordination number of Pt–O in 0.1Pt/C12A7 also exhibits high catalytic activity towards hydrogenation reactions, which is in good agreement with that work.³

As we mentioned in Question 1, ca. 20% fitting error is acceptable in EXAFS analysis. The coordination number fitted from the EXAFS results was an average number of all probable Pt–O coordination manners. Therefore, Pt–O with CN~3.34 of 0.1Pt/C12A7 and CN~2.52 of 0.1Pt/C12A7-R600 can be considered as almost the same coordination environment. To confirm this point, we additionally measured the FTIR spectra of CO adsorption on 0.1Pt/C12A7 and 0.1Pt/C12A7-R600. As shown in Fig. S19, the band at 2070 cm^{-1} of linearly adsorbed CO remained unchanged on 0.1Pt/C12A7 and 0.1Pt/C12A7-R600 respectively, indicating the same coordination environment of single Pt atoms in both samples. Accordingly, it is consistent that the catalytic performance of the two samples gave similar hydrogenation activity (Fig. 4a), which further demonstrated the similar Pt coordination number in 0.1Pt/C12A7 and 0.1Pt/C12A7-R600.

Fig. S19 FTIR spectra of CO adsorption on 0.1Pt/C12A7, 0.1Pt/C12A7-R600, 0.1acacPt/C12A7, 0.1Pt/ Al_2O_3 and 0.1Pt/CaO, respectively.

In order to make these points more clear, we added the following explanation to the revised main-text.

(Revised manuscript, page 10, line 3–5 and page 15, line 28–30)

Here, Pt–O with CN~3.34 of 0.1Pt/C12A7 and CN~2.52 of 0.1Pt/C12A7-R600 can be considered as almost the same coordination environment by taking into account the ca. 20% fitting error in EXAFS analysis.

Here, the hydrogen activation process should also be promoted by the Pt single atoms with low coordination number of Pt–O in the surface cavities of C12A7, which is in good agreement with Pt₁/Fe₂O₃.

3. Please show how you calculated the TOF value. For example, by using the data in Table S2, the TOF value of the 0.1Pt/C12A7 was only 8125 h⁻¹ (what I calculated), rather than 25772 h⁻¹. Moreover, the comparison with literature (Table S3-S5) should be at similar reaction conditions. For the fair comparison, you should use the reaction data at 2 h rather than the initial several minutes. For this reason, your claim that “surpassing the catalytic performance of all previously reported Pt-based heterogeneous catalysts under similar conditions” (line 293-294) needs to be revised.

R: First of all, we would like to appreciate the referee for attracting our attention to this issue. The method for TOF calculation was briefly described in the experimental section “The TOF value was separately measured by keeping the substrate conversion below 20% and the TOF calculation was based on the total Pt loading in the catalyst”. Here, the value was calculated in the format of $\text{mol}_{4\text{-chloroaniline}} \text{mol}^{-1}_{\text{total Pt}} \text{h}^{-1}_{\text{reaction time}}$. The amount of metal is based on the moles of total noble metal involved. We added this additional description into the experimental section. The TOF value of 25772 h⁻¹ was calculated from the reaction rate at 5 mins under 60 °C which can be found in Fig. S26. The TOF value of 8125 h⁻¹ supposed by the referee can be obtained through 2h reaction at 60 °C.

As suggested by the referee, the TOFs based on 2 hrs reaction at 60 °C were also calculated and added into modified Table S4-6, which are also several times higher than those of reported different Pt-based heterogeneous catalysts under similar conditions. It is worth noting that the real TOFs values should be calculated based on the kinetic reaction rates at low conversion. Because the reaction rate tends to decrease due to the progressively lowered chance of a collision between substrates. So TOFs based on the reaction rates at initial several minutes are much more reasonable to illustrate the true catalytic activity. Therefore, the TOFs calculated based on 5mins reactions is also kept in Table S4-6.

4. In table S2, for the samples with increased Pt loadings, the activity might be higher. In this case, the reaction time of 2 h might be so long that the side reactions took place. For the selectivity comparison at similar conversion, the reaction time might be varied depending on the Pt loading.

R: We thank the referee for attracting our attention to this issue. We agree that the shorter reaction time is needed to confirm the selectivity on high Pt loading samples. To eliminating the possibility that the poor catalytic performance of high Pt loading amount samples was arose from the over-reaction, we additionally measured the activity and selectivity of 1.0Pt/C12A7 and 2.0Pt/C12A7 at 0.5h with much lower conversion (54.7% and 69.2%, respectively). Indeed, both 1.0Pt/C12A7 and 2.0Pt/C12A7 showed similar selectivity compared to those of 2h reaction time (shown in modified Table S3, entries 6 and 8). Therefore, we can conclude that the larger particle size of Pt NPs accounts for poor selectivity, which may be due to the disparate adsorption behavior for the substrates.

Table S3 Chemoselective hydrogenation of 4-chloronitrobenzene on different Pt/C12A7 catalysts

Entry	Catalyst	T (°C)	t (h)	Conv. (%)	Select. (%)		
					2	3	others
1	0.1Pt/C12A7	60	2	99.9	99.9	0	0
2	0.1Pt/C12A7	25	9	99.9	99.9	0	0
3	0.3Pt/C12A7	60	2	99.9	99.3	0.6	0
4	0.5Pt/C12A7	60	2	99.9	98.9	1.0	0
5	1.0Pt/C12A7	60	2	99.9	82.3	12.5	5.1
6	1.0Pt/C12A7	60	0.5	54.7	75.6	16.4	7.9
7	2.0Pt/C12A7	60	2	99.9	77.9	12.9	9.1
8	2.0Pt/C12A7	60	0.5	69.2	79.2	13.6	7.1
9	C12A7	60	9	---	---	---	---
10	Blank	60	9	---	---	---	---

Reaction condition: 0.5 mmol substrate, 5 mg catalyst, 5 ml methanol, 0.5 MPa H₂. Conversion (Conv.) and selectivity (Select.) were determined by GC and GCMS using n-hexadecane as an internal standard.

5. Grammar errors:

Line 69, “is highly desirable for the preparation of and the long-term operation of SACs.”

R: We appreciate the referee for his/her carefulness. Here we deleted the word “of” and the sentence was modified was “is highly desirable for the preparation and the long-term operation of SACs.”

Line 204, “The coordination number around Pt is 1.45 (Table 1).” The description is not accurate because there are two shells in Table 1, Pt-O and Pt-Pt. Please indicate which shell. Moreover, the figure in Table 1 is 1.43, rather than 1.45.

R: We would like to appreciate the referee for pointing out this issue. Here, we modified this sentence to “The Pt-O coordination number is 1.43 (Table 1).”

Reference

1. Zhang, B. et al. Stabilizing a platinum₁ single-atom catalyst on supported phosphomolybdic acid without compromising hydrogenation activity. *Angew. Chem. Int. Ed.* **55**, 8319–8323 (2016).
2. Chen, Y. et al. Isolated Single Iron Atoms Anchored on N-Doped Porous Carbon as an Efficient Electrocatalyst for the Oxygen Reduction Reaction. *Angew. Chem. Int. Ed.* **56**, 6937–6941 (2017).
3. Ren, Y. et al. Unraveling the coordination structure-performance relationship in Pt₁/Fe₂O₃ single-atom catalyst. *Nat. Commun.* **10**, 4500 (2019).

Reviewer #2 (Remarks to the Author):

1) Recently, the relationship between the metal-support strength and single Pt atom stability/activity has been systematically studied for the first time (JACS, 2019, 141, 8185-8197.) Two major findings in the paper include 1) the least adsorption energy to obtain sintering resistant Pt SACs is 5.50 eV, exactly matching the cohesive energy of bulk Pt metal; and 2) stable Pt SACs may not necessarily be less active due to several competing factors. The current study is a nice example to illustrate the correctness of the earlier study. First, the adsorption energy of Pt in the cavity is -9 eV, far more than the cohesive energy of bulk Pt. Also, such stable Pt SACs are very active in nitro reduction reaction. Therefore, credit should be given to the JACS paper where quantitative correlation between Pt-support interaction and stability/hydrogenation activity have already been given.

R: We would like to appreciate the referee for reviewing our manuscript. We are also grateful to the referee for recommending an excellent work of Pt single atom catalyst.¹ In that work, a series of polyoxometalate-supported Pt single atom catalysts were used as models to investigate the effect of metal-support interaction on catalytic activity and stability through both experimental and theoretical approaches. Indeed, the reported Pt sintering resistant energy and the Pt-support interaction are instructive for the designed synthesis and catalytic application for SACs. We are also very happy that our DFT calculation and experimental results further illustrate the correctness of the JACS paper. Therefore, we emphasized the importance of this work and also cited it in modified manuscript.¹

2) The major issue with the scientific part is the lack of evidence for the hydride mechanism. A pronounced KIE effect (>4) only suggest the rate-determining step in the Pt SAC catalyst is different from the two other samples. It can not be used as a proof for the existence of the hydride mechanism. To the minimum, the authors should conduct further DFT calculations to show that the hydride might be the preferred pathway.

R: We would like to appreciate the referee for attracting our attention to this issue. Indeed, the KIE result itself can't be able to determine the RDS in hydride mechanism. However, as the loaded Pt single atoms are composed of Pt^{δ+}, heterolytic dissociation of hydrogen promote to

form $\text{O-H}^{\delta+}$ and $\text{Pt-H}^{\delta-}$ species, which is proved in other single atoms catalysts for hydrogenation reactions.²⁻⁴ Here, we also supplied the FTIR spectroscopy results (Fig. 4c and Figs. S32, S33) that the formation of $\text{O-H}^{\delta+}$ was only observed on 0.1Pt/C12A7, indicating both $\text{H}^{\delta+}$ and $\text{H}^{\delta-}$ were generated at the Pt-O interface which can be considered as a frustrated Lewis pair (FLP) site.² In combination with different KIE effect and FTIR results, a hydride mechanism can be proposed in our reaction system.

3) There is a lack of reaction kinetic information. The authors select conditions where near 100% conversions are achieved, and therefore hide the intrinsic kinetic data. The reaction order for both nitrocompounds and H_2 should be measured.

R: We would like to appreciate the referee for attracting our attention to this issue. As suggested by the referee, we additionally investigated the dependency of the initial reaction rates on the partial pressures of H_2 and nitroarenes concentration. As shown in Fig. S34, the reaction rates over 0.1Pt/C12A7 are more sensitive to the hydrogen pressure compared with the 4-chloronitrobenzene concentration, which results in kinetic reaction orders of $\alpha(\text{H}_2) = 0.94$ and $\beta(4\text{-chloronitrobenzene}) = 0.36$, respectively. Also, we would not like to hide intrinsic kinetic data since the comparison of TOFs values and apparent activation energy (Fig. 4) were all calculated from the kinetic reaction rates at low conversion levels.

Fig. S34 Dependence of reaction rate on partial pressures of H_2 and 4-chloronitrobenzene concentration.

In order to make these points more clearly, we added the following explanation to the revised main-text.

(Revised manuscript, page 15, line 15–20)

In addition, Fig. S34 shows the dependency of the initial reaction rates on the partial pressures of H₂ and nitroarenes concentration. The reaction rates over 0.1Pt/C12A7 are more sensitive to the hydrogen pressure compared with the 4-chloronitrobenzene concentration, which results in kinetic reaction orders of $\alpha(\text{H}_2) = 0.94$ and $\beta(4\text{-chloronitrobenzene}) = 0.36$, respectively.

4) The CO adsorption on all Pt samples should be provided. This provides additional clues whether Pt species are truly fully atomically dispersed. HRTEM only tells a tiny fraction of the samples while XAS is not sensitive enough to detect a small amount of NPs (for example, less than 10%).

R: We would like to appreciate the referee for attracting our attention to this issue. As suggested by the referee, we additionally measured the FTIR spectra of CO adsorption as shown in Fig. S19. For 0.1Pt/C12A7 and 0.1Pt/C12A7-R600, the band at 2070 cm⁻¹ is ascribed to CO linearly adsorbed on Pt^{δ+}, confirming that the Pt^{δ+} single atoms in 0.1Pt/C12A7 and 0.1Pt/C12A7-R600 are well isolated, which is consistent with the reported CO adsorption on Pt₁/FeO_x.⁵ For 0.1acacPt/C12A7, 0.1Pt/Al₂O₃ and 0.1Pt/CaO, the bands at 2040 cm⁻¹ is ascribed to the linearly bonded CO on Pt⁰ sites. Moreover, the band at 1850 cm⁻¹ for the bridge-bonded CO on neighboring Pt atoms is observed for 0.1Pt/Al₂O₃ and 0.1Pt/CaO.⁵ In combination with HAADF-STEM images, Pt K-edge EXAFS spectra and FTIR spectra of CO adsorption, it is reasonable to conclude that Pt^{δ+} species are atomically dispersed on the surface of C12A7 with exceptional high thermal stability.

Fig. S19 FTIR spectra of CO adsorption on 0.1Pt/C12A7, 0.1Pt/C12A7-R600, 0.1acacPt/C12A7, 0.1Pt/Al₂O₃ and 0.1Pt/CaO, respectively.

In order to make these points more clearly, we added the following explanation to the revised main-text. Detailed experimental procedures were also added into the revised experimental section.

(Revised manuscript, page 10, line 10–18)

Fourier-transform infrared (FTIR) spectroscopy of CO adsorption behavior were also studied to verify the Pt atoms on C12A7. The only adsorption peak with band at 2070 cm⁻¹ ascribed to CO linearly adsorbed on Pt^{δ+} were detected on both 0.1Pt/C12A7 and 0.1Pt/C12A7-R600 (Fig. S19). While the signals ascribed to CO adsorption on Pt⁰ sites and bridge sites of Pt clusters or NPs can be observed on 0.1acacPt/C12A7, 0.1Pt/Al₂O₃ and 0.1Pt/CaO, respectively (Fig. S19), but not on 0.1Pt/C12A7 and 0.1Pt/C12A7-R600, which further confirmed the atomic dispersion of single atom Pt^{δ+} species on the surface of C12A7 with exceptional high thermal stability.

5) What also puzzles the reviewer, is that while the major advantage of the catalyst is the high temperature stability under reducing condition, the catalyst is only used for nitro group reduction reactions at 60 degree C. Why not trying more demanding reactions that require 200 degree C and above, under H₂ atmosphere?

R: In response to the referee's suggestion, we additionally performed the CO oxidation reaction under relative higher temperature. As shown in Fig. S30a, 0.1Pt/C12A7 and 0.1Pt/C12A7-R600 exhibited similar trends in terms of conversion and reached 100% CO conversion at around 225 °C. Moreover, the catalytic stability of 0.1Pt/C12A7 catalyst was also investigated in a continuous flow of the reactant gas at 175 °C and 250 °C for 80 h respectively (Fig. S30b). In both stability experiments, no obvious deactivation occurred. These results further illustrated the high temperature stability of 0.1Pt/C12A7 single atom catalyst even under oxidative condition.

Fig. S30 Evaluation of 0.1Pt/C12A7 catalysts in CO oxidation reaction. **a.** Conversion of CO from 50 to 230 °C over 0.1Pt/C12A7 and 0.1Pt/C12A7-R600 respectively. **b.** Time course for CO conversion at 175 °C and 250 °C over 0.1Pt/C12A7 catalyst.

In order to clarify this point, we added these experiment results into the supporting information and the following explanation to the revised main-text. Detailed experimental procedures were also added into the revised experimental section.

To further confirm the catalytic stability of 0.1Pt/C12A7 at high temperature, CO oxidation reaction was also investigated. As shown in Fig. S30a, 0.1Pt/C12A7 and 0.1Pt/C12A7-R600 exhibited similar trends in terms of conversion and reached 100% CO conversion at around 225 °C. Moreover, the catalytic stability of 0.1Pt/C12A7 catalyst was also investigated in a continuous flow of the reactant gas at 175 °C and 250 °C for 80 h respectively (Fig. S30b). In both stability experiments, no obvious deactivation occurred. These results further illustrated the high temperature stability of 0.1Pt/C12A7 SACs even under oxidative condition.

6) Based on Figure 1, the C12A7 material should be able to support single-atom Pt in a very high loading, if each site can indeed stabilize one Pt atom. In reality, a loading of 0.1 wt% was used. How do we understand this? If only stable Pt SAC could only be prepared under such a low loading, the reviewer has reasons to doubt that Pt may not locate on C12A7 as the authors suggested.

R: We would like to appreciate the referee for attracting our attention to this issue. Here, Fig. 1 only shows a schematic diagram to explain how the Pt precursors can be fixed on the surface cavities of C12A7. Actually, the number of surface cavities is limited. From CO₂-TPD experiment in Fig. 3, the amount of adsorbed CO₂ molecules can be estimated through the calibration of CO₂ standard gas by mass spectrometry. According to our previous work of CO₂ adsorption on C12A7:e⁻, each surface truncated nanocage only corresponds to one CO₂ molecule (Fig. 3f inset), which can be used to determine the concentration of surface truncated nanocages of C12A7. As shown in Table S2, the amount of surface truncated nanocage is ca. 9.1 μmol/g, which means the upper limit of the single atomic Pt is 1.8 mg_{Pt}/g_{catalyst} (~0.18wt%). Therefore, the real Pt loading amount in our experiment is ca. 0.12wt% through ICP analysis, which is close to the limited value.

Table S2 TPD-CO₂ profile of various compared samples

Sample	Surface area (m ² /g)	Amount of basic site (μmol of CO ₂ /g)				Surface truncated nanocages
		Al-O-Al	Ca-O-Al	^a OH ⁻	^b Ca-O-Ca	
C12A7	54	0.4	1.6	0.9	8.8	9.1
0.1Pt/C12A7	52	0.4	1.8	1.0	9.2	3.3

^aInitial extra-framework ion that merged with the cage wall in the surface truncated nanocages.

^bSmall amount of uncrystallized CaO species under low temperature (600 °C) calcination of hydrothermal obtained C12A7 precursor.

In order to clarify this point, we added these experiment results and the following explanation into the revised supporting information.

(Revised manuscript, page 12, line 11–17)

Here, the amount of adsorbed CO₂ molecules can be estimated through the calibration of CO₂ standard gas by mass spectrometry. According to our previous work of CO₂ adsorption on C12A7:e⁻, each surface truncated nanocage only corresponds to one CO₂ molecule (Fig. 3f inset), which can be used to determine the concentration of surface truncated nanocages of C12A7. As shown in Table S2, the amount of surface truncated nanocage is ca. 9.1 μmol/g, which means the upper limit of the single atomic Pt is 1.8 mg_{Pt}/g_{catalyst} (~0.18wt%).

7) It would be nice that at least the authors comment whether their support is able to anchor metal atoms beyond Pt.

R: We would like to appreciate the referee for his nice suggestion. Here, in order to confirm whether the trapping effect of the unique surface cavities of C12A7 also applicable to other noble metals, we tried the similar strategy as Pt to prepare other single atom noble metal loaded on C12A7. As shown in the high-resolution HAADF-STEM images (Fig. S8), isolated single atoms such as Ru and Rh dispersed on the C12A7 surface are clearly visible, which can be

further confirmed by corresponding EDX and EELs analysis (Figs. S9, S10). These results demonstrated that the single atom trapping effect of C12A7 here is general toward a large variety of SACs.

Fig. S8 HAADF-STEM images of (a) 0.1Ru/C12A7 and (b) 0.1Rh/C12A7 single-atom structures. Single atoms marked in yellow circles dispersed on the C12A7 support.

In order to clarify this point, we added these experiment results into the supporting information and the following explanation to the revised main-text.

(Revised manuscript, page 8–9, line 1–5)

Inspired by this strategy, we wondered whether the trapping effect of the unique surface cavities of C12A7 also applicable to other noble metals. As shown in the high-resolution HAADF-STEM images (Fig. S8), isolated single atoms such as Ru and Rh dispersed on the C12A7 surface are clearly visible, which can be further confirmed by corresponding EDX and EELS analysis (Fig. S9, S10). These results demonstrated that the single atom trapping effect of C12A7 here is general toward a large variety of SACs.

8) Finally, the single-atom catalysis research is a fast growing area. Some relevant papers, such as Pt SAC for nitro group reduction (e.g., *Angew. Chem. Int. Ed.*, 2016, 55, 8319–8323.) and review articles on enhancing the stability of SACs (e.g., *Adv. Mater.*, 2018, 30, 1802304.) should be cited.

R: We would like to appreciate the referee for sharing some other relevant Pt SAC works.^{6,7} We are also happy to cite these two excellent papers in the modified manuscript.

Reference

1. Zhang, B. et al. Atomically Dispersed Pt₁-Polyoxometalate Catalysts: How Does Metal-Support Interaction Affect Stability and Hydrogenation Activity. *J. Am. Chem. Soc.* **141**, 8185–8197 (2019).
2. Liu, P. et al. Photochemical route for synthesizing atomically dispersed palladium catalysts. *Science* **352**, 797–801 (2016).
3. Liu, W. et al. A durable nickel single-atom catalyst for hydrogenation reactions and cellulose valorization under harsh conditions. *Angew. Chem. Int. Ed.* **57**, 7071–7075 (2018).
4. Bai, L. et al. Explaining the Size Dependence in Platinum-Nanoparticle-Catalyzed Hydrogenation Reactions. *Angew. Chem. Int. Ed.* **55**, 15656–15661 (2016).

5. Qiao, B. et al. Single-atom catalysis of CO oxidation using Pt₁/FeO_x. *Nat. Chem.* **3**, 634–641 (2011).
6. Zhang, B. et al. Stabilizing a platinum₁ single-atom catalyst on supported phosphomolybdic acid without compromising hydrogenation activity. *Angew. Chem. Int. Ed.* **55**, 8319–8323 (2016).
7. Hülsey, M. J., Zhang, J. & Yan, N. Harnessing the wisdom in colloidal chemistry to make stable single-atom catalysts. *Adv. Mater.* **30**, 1802304 (2018).

Reviewer #3 (Remarks to the Author):

The list of classes of SAC is too short. Isolated atoms on other metals and on carbons are by far one of the most investigated.

R: We would like to appreciate the referee for pointing out this issue. Here, in order to extend our single atom strategy to other noble metal, we tried the similar route as Pt to prepare other single atom noble metal loaded on C12A7. As shown in the high-resolution HAADF-STEM images (Fig. S8), isolated single atoms such as Ru and Rh dispersed on the C12A7 surface are clearly visible, which can be further confirmed by corresponding EDX and EELS analysis (Figs. S9, S10). These results demonstrated that the single atom trap effect of C12A7 here is general toward a large variety of SACs.

Fig. S8 HAADF-STEM images of (a) 0.1Ru/C12A7 and (b) 0.1Rh/C12A7 single-atom structures. Single atoms marked in yellow circles dispersed on the C12A7 support.

In order to clarify this point, we added these experiment results into the supporting information and the following explanation to the revised main-text.

(Revised manuscript, page 8–9, line 1–5)

Inspired by this strategy, we wondered whether the trap effect of the unique surface cavities of C12A7 also applicable to other noble metals. As shown in the high-resolution HAADF-STEM images (Fig. S8), isolated single atoms such as Ru and Rh dispersed on the C12A7 surface are clearly visible, which can be further confirmed by corresponding EDX/EEELs and ICP-OES analysis (Figs. S9, S10). These results demonstrated that the single atom trap effect of C12A7 here is general toward a large variety of SACs.

The use of mixed units (eV and kJ/mol) should be avoided.

R: We appreciate the referee for pointing out this issue. We unified the unit into eV.

Energies of -10 eV like Pt adsorption basically means that the zero reference is completely misplaced. A better reference needs to be chosen. Particularly, when Figure 1 reports PtCl₂- as reference.

R: We appreciate the referee for attracting our attention to this issue. In our DFT calculation, one Pt single atom and bare C12A7 substrate was chosen as the zero reference and the adsorption energy was calculated by using the equation as following:

$$E_{\text{ads}} = E_{\text{total}}(\text{Pt}_n/\text{C12A7}) - nE(\text{Pt}) - E_{\text{total}}(\text{surface})$$

We added this modification into the revised experimental section.

Here, PtCl_2^- is not a suitable as reference, because Cl^- ions will incorporate into the sub-cages of C12A7 during the thermal reduction process, which will be discussed later.

The speciation has shown to be crucial for the reactivity (see ACIE and Nature Comm. Perez-Ramirez 2019). Dimerization energies need to be calculated much before aggregation can be discarded only on terms of energies. Pt-Pt distances can be larger than 2-3 Angstrom found in the experiments (see discussion ACIE 2019).

R: We appreciate the referee for pointing out this issue. Indeed, the speciation plays a vital role in catalytic activity as well as selectivity according to the literature.^{1,2} As suggested by the referee, DFT calculations were also conducted to assess the dimerization and trimerization energies respectively.

The dimerization and trimerization energy per Pt atom in the ensemble (E_{form}) of the Pt/C12A7 catalysts were calculated to be 2.15 eV and 2.61 eV for Pt₂/C12A7 and Pt₃/C12A7 respectively (Fig. S1), which are prohibitively high so that the aggregation of Pt atoms to form dimers and trimers is not favored. Also the adsorption energy of -7.38 eV/Pt and -6.92 eV/Pt of Pt dimer and trimer on C12A7 (001) further demonstrate that the interaction of Pt single atoms on surface cavities of C12A7 is stronger than those of Pt dimers and trimers. These results well demonstrated that unique surface cavity structures of C12A7 with inner diameters of about 0.4 nm is just the right size for trapping one Pt single atom, which is in good agreement with our experimental results.

Fig. S1 DFT optimized adsorption models of Pt dimer and trimer on the (001) surface of C12A7. The adsorption energy of Pt dimer and trimer is -7.38 eV/Pt and -6.92 eV/Pt respectively. Meanwhile, the dimerization and trimerization energy per Pt atoms is also calculated from the Pt ensemble effect, showing 2.15 eV/Pt and 2.61 eV/Pt respectively, which are prohibitively high so that the dimerization and trimerization can hardly occur.

In order to clarify this point, we added these experiment results into the supporting information and the following explanation to the revised main-text. Detailed calculation procedures were also added into the revised experimental section.

(Revised manuscript, page 5, line 10–12)

Moreover, the formation of Pt dimers and trimers can also be excluded due to the high dimerization and trimerization energy of 2.15 eV/Pt and 2.61 eV/Pt, respectively (Fig. S1).

The description of the atoms needs some electronic density analysis. Bader charges, DOS, XPS shifts all of these characterize properly the structure of the atoms.

R: As suggested by the referee, we have performed Bader charge analysis. The charge states of the most stable adsorbed Pt single atom on the C12A7 (001) surface was estimated to be +0.15, indicating an obviously electron transfer from Pt to neighbor O, which is consistent with the XAFS and CO-FTIR results. From the projected d-band densities of states (DOS), we found an obvious up-shift of the d-band center for Pt single atom on C12A7 (001) to -2.04 eV, whereas pure Pt (111) is -2.59 eV (Fig. S37). Since the polarized charge would effectively change the d-band states of Pt atom, thus the d-band states of the transition metal have a significant influence on the interactions between the transition metal and the adsorbed reactant. When the d-band states of transition metal shift toward the Fermi level, the antibonding states of transition metal will be pushed above the Fermi level and reduce the Pauli repulsion, which will make stronger interactions between the transition metal and the adsorbed reactant.³ In our case, the Pt atoms with up-shift d-band states are much nearer to the Fermi level, resulting much stronger adsorption capability for the reactants such as H_2 and nitroarenes, which is consistent with the TPD-DRIFT experimental results (Fig. S35).

Fig. S37 Projected Pt *d*-band densities of states (DOS). The *d*-band center (ϵ_d) is at -2.59 , and -2.04 eV for Pt (111) surface and Pt₁/C12A7 respectively.

As suggested by the referee, the XPS measurement was performed to reveal Pt valence state. Since Pt 4f spectra overlapped with Al 2p signal, here we show the Pt 4d_{5/2} spectra of 0.1Pt/C12A7 and 2.0Pt/C12A7 (Fig. R1). Notably, there is almost no Pt 4d_{5/2} signal over 0.1Pt/C12A7 due to the detection limit of the XPS instrument for the extremely low concentration of surface Pt species. In contrast, a single Pt 4d_{5/2} binding energy of 316.1 eV, corresponding to Pt⁰, was observed for 2.0Pt/C12A7 (Fig. R1).

To further characterize the properties of surface Pt atoms, we additionally collected the CO-FTIR spectra. As shown in Fig. S19, the band at 2070 cm⁻¹ is ascribed to CO linearly adsorbed on Pt^{δ+}, confirming that the single Pt atoms in 0.1Pt/C12A7 and 0.1Pt/C12A7-R600 are well isolated, which is consistent with the CO adsorption of reported Pt₁/FeO_x.⁴ These results well demonstrated that single atom Pt^{δ+} species are atomically dispersed on the surface of C12A7 with exceptional high thermal stability, which is in good agreement with our XAFS results.

Fig. R1 XPS Pt 4d_{5/2} spectra for the as-prepared 0.1Pt/C12A7 and 2.0Pt/C12A7 catalysts.

Fig. S19 FTIR spectra of CO adsorption on 0.1Pt/C12A7, 0.1Pt/C12A7-R600, 0.1acacPt/C12A7, 0.1Pt/Al₂O₃ and 0.1Pt/CaO, respectively.

In order to clarify this point, we added these experiment results into the supporting information and the following explanation to the revised main-text.

(Revised manuscript, page 7, line 25–29, page 16, line 16–22, page 10, line 10–18)

The normalized X-ray absorption near-edge structure (XANES) spectra in Fig. S4 shows that the peak above the edge (white-line) for 0.1Pt/C12A7 is located between those for Pt foil and PtO₂, implying a slightly positively charged Pt^{δ+} rather than Pt⁰, which agrees with the Bader charge analysis of +0.15 of adsorbed Pt single atom on the C12A7 (001) surface.

In addition, the partial density of states (PDOS) of Pt atom on surface of C12A7 were also calculated in Fig. S37. It is observed that the Pt atoms with up-shift d-band states are much close to the Fermi level, resulting in much stronger adsorption capability for the reactants such as H₂ and

nitroarenes. Both experimental TPD-DRIFT and calculated d-band center shift are in agreement with the superior catalytic activity of Pt single atoms loaded C12A7 catalysts.

Fourier-transform infrared (FTIR) spectroscopy of CO adsorption behavior were also studied to verify the Pt atoms on C12A7. The only adsorption peak with band at 2070 cm^{-1} ascribed to CO linearly adsorbed on $\text{Pt}^{\delta+}$ were detected on both 0.1Pt/C12A7 and 0.1Pt/C12A7-R600 (Fig. S19). While the signals ascribed to CO adsorption on Pt^0 sites and bridge sites of Pt clusters or NPs can be observed on 0.1Pt/C12A7, 0.1Pt/Al₂O₃ and 0.1Pt/CaO, respectively (Figs. S19), but not on 0.1Pt/C12A7 and 0.1Pt/C12A7-R600, which further confirmed the atomic dispersion of single atom $\text{Pt}^{\delta+}$ species on the surface of C12A7 with exceptional high thermal stability.

How do the authors consider the elimination of Cl from the precursor?

R: We would like to appreciate the referee for attracting our attention to this issue. Here, the Cl species could not be eliminated during the preparation process. Instead, due to the unique anion-exchange properties of C12A7, Cl⁻ ions can replace the OH⁻ ions and be stored in the sub-cages of C12A7. In this work, we also measured the XPS spectrum to further confirm the existence of Cl⁻ species in 0.1Pt/C12A7 catalyst. As shown in Fig. S3, the binding energy of Cl 2p was observed at ca. 197.8 eV, revealing the existence of Cl⁻ in the C12A7. Most importantly, the subsurface resident Cl⁻ gave negligible effect on the catalytic performance of nitroarenes hydrogenation reactions, which also agrees with our previous work of C12A7 based Cl-tolerant catalyst for ammonia synthesis.⁵

Fig. S3 XPS Cl 2p spectrum for the as-prepared 0.1Pt/C12A7 catalyst.

In order to clarify this point, we added these experiment results into the supporting information and the following explanation to the revised main-text.

(Revised manuscript, page 7, line 8–11)

XPS measurement confirmed the existence of Cl^- ions (Fig. S3), indicating that Cl^- ions replaced the OH^- ions in the sub-cages of C12A7 and were stored in them during the reduction process.

The hydrogenation of nitrobenzene has been reported in the literature supported by detailed calculations (Perez-Ramirez, ChemCatChem, ACS Catal). The authors can map their discussion to the previous description in the literature. The full reaction can be calculated to illustrate the enhancement.

R: We would like to appreciate the referee for sharing some other excellent works with respect to hydrogenation of nitroaromatics.^{6,7} Both works are based on the continuous gas-phase catalytic reaction process to study the hydrogenation of nitroarenes over Cu-based and Pt-based catalysts. In the former work, Cu-containing hydrotalcite-derived catalyst was firstly reported in the

catalytic hydrogenation nitroarenes. Moreover, the incorporation of Ni resulted in a significant increase in hydrogenation rate with 100% selectivity to p-chloroaniline. In the later one, carbon supported hybrid Pt nanoparticles catalyst was developed for flow hydrogenation of functionalized nitroaromatics. In addition to its excellent activity, the reaction mechanism was firstly analyzed systematically by DFT calculation. In combination with experimental and calculation results, promoted H₂ activation was achieved on this ligand-modified catalyst, which accounts for the superior catalytic activity. Meanwhile, Pt ensembles isolated by the ligands could control the adsorption geometry of the reactant and product intermediates, which is conducive to obtain high selectivity of the target products.

In our work, the unique surface cavity structure of C12A7 can also be considered as a ligand to trap the Pt single atom, preventing the sintering and aggregation problem. The isolated Pt^{δ+} and the adjacent O atom can be considered as a frustrated Lewis pair (FLP) site, benefit for the heterolytic cleavage of adsorbed H₂, allowing better hydrogenation of polar unsaturated bonds such as nitro groups. Most importantly, under the same loading amount, single Pt SACs can provide much more efficient active sites for H₂ dissociation, which accelerates the hydrogenation reaction and delivers higher catalytic activity. From the calculated PDOS results, the Pt atoms with up-shift d-band states are much close to the Fermi level, resulting in much stronger adsorption capability for the reactants such as H₂ and nitroarenes. Meanwhile, the TPD-DRIFT experimental results also reveal the strong adsorption of nitro groups, which is favored for the activation of the nitro groups.

What is the reaction order of H₂? and of nitrobenzene? This can help in the investigation.

R: We would like to appreciate the referee for attracting our attention to this issue. As suggested by the referee, we additionally investigated the dependency of the initial reaction rates on the partial pressure of H₂ and nitroarenes concentration. As shown in Fig. S34, the reaction rates over 0.1Pt/C12A7 are more sensitive to the hydrogen pressure compared with the 4-chloronitrobenzene concentration, which results in kinetic reaction orders of $\alpha(\text{H}_2) = 0.94$ and $\beta(4\text{-chloronitrobenzene}) = 0.36$, respectively.

Fig. S34 Dependence of reaction rate on partial pressures of H₂ and 4-chloronitrobenzene concentration.

In order to clarify this point, we added these experiment results into the supporting information and the following explanation to the revised main-text.

(Revised manuscript, page 15, line 15–20)

In addition, Fig. S34 shows the dependency of the initial reaction rates on the partial pressures of H₂ and nitroarenes concentration. The reaction rates over 0.1Pt/C12A7 are more sensitive to the hydrogen pressure compared with the 4-chloronitrobenzene concentration, which results in kinetic reaction orders of $\alpha(\text{H}_2) = 0.94$ and $\beta(4\text{-chloronitrobenzene}) = 0.36$, respectively.

Charges on atoms can change depending on the oxide (Daelman Nature Mat. 2019) are the authors sure that this does not contribute to their work.

R: We would like to appreciate the referee for sharing this excellent work with respect to the dynamic charge state of Pt single atom species on ceria.⁸ In that work, Prof. López et al. firstly discovered the co-existed several charge states of loaded Pt single atoms on CeO₂ (100) surface

through DFT calculation and first-principle molecular dynamics, which is closely related to the catalytic activity of single atom catalysts. The co-existence of several coordination and oxidation states of Pt originated from the dynamic charge transfer between the metal and the oxide support, which is assisted by phonons. Based on this discovery, CO oxidation was chosen as a model reaction to investigate the relationship between variable oxidation states of Pt single atom and their catalytic activity. In their conclusion, the single Pt^+ atoms with short lifetime was confirmed to be the most active species.

In our work, the DFT calculation as shown in Fig. 1a, Pt single atoms can be stabilized by various surface oxygen sites of C12A7 (001) with four kinds of Pt-O configurations, which should also be different electronic states of the Pt species. Here, we can't rule out the co-existence of these different coordination states only based on the DFT calculation. However, the XAFS, CO_2 -TPD and CO-FTIR experimental results show that single atom $\text{Pt}^{\delta+}$ species are atomically dispersed in the surface cavities of C12A7, which is consistent with the most stable adsorption configurations of position 3 (Fig. 1b). Most importantly, the polarized charge between Pt-O bond induced positive charged Pt single atoms ($\text{Pt}^{\delta+}$) can be confirmed by both experiments and DFT calculations, exhibiting superior catalytic activity towards hydrogenation reaction of nitroarenes, which is also in good agreement with the conclusion of the Nature Material paper.

Reference

1. Vorobyeva, E. et al. Atom-by-atom resolution of structure–function relations over low-nuclearity metal catalysts. *Angew. Chem. Int. Ed.* **58**, 8724–8729 (2019).
2. Frei, M. S. et al. Atomic-scale engineering of indium oxide promotion by palladium for methanol production via CO_2 hydrogenation. *Nat. Commun.* **10**, 3377 (2019).
3. Xiao, J. & Frauenheim, T. Theoretical insights into CO_2 activation and reduction on the Ag(111) monolayer supported on a ZnO(0001) substrate. *J. Phys. Chem. C* **117**, 1804–1808 (2013).

4. Qiao, B. et al. Single-atom catalysis of CO oxidation using Pt₁/FeO_x. *Nat. Chem.* **3**, 634–641 (2011).
5. Li, J. et al. Chlorine-tolerant ruthenium catalyst derived using the unique anion-exchange properties of 12CaO·7Al₂O₃ for ammonia synthesis. *ChemCatChem* **9**, 3078–3083 (2017).
6. Cárdenas-Lizana, F., Bridier, B., Shin, C. C. K., Pérez-Ramírez, J. & Kiwi-Minsker, L. Promotional effect of Ni in the selective gas-phase hydrogenation of chloronitrobenzene over Cu-based catalysts. *ChemCatChem* **4**, 668–673 (2012).
7. Vilé, G., Almora-Barrios, N. López, N. & Pérez-Ramírez, J. Structure and Reactivity of Supported Hybrid Platinum Nanoparticles for the Flow Hydrogenation of Functionalized Nitroaromatics. *ACS Catal.* **5**, 3767–6778 (2015).
8. Daelman, N., Capdevila-Cortada, M. & López, N. Dynamic charge and oxidation state of Pt/CeO₂ single-atom catalysts. *Nat. Mater.* (2019), DOI:doi.org/10.1038/s41563-019-0444-y.

REVIEWERS' COMMENTS:

Reviewer #1 (Remarks to the Author):

I am satisfied with the authors' responses as well as additional data to improve the paper. I now recommend its publication.

Reviewer #2 (Remarks to the Author):

The authors have addressed most of my comments. However, there is room for improvement for the newly added CO-IR data (see below). Once that is addressed, I would be pleased to recommend the acceptance of the paper.

The author used CO-DRIFT to demonstrate the single-atom identify of Pt. The spectra were provided in Figure S19. But a bit more discussion should be added to the text to enhance scientific level.

First, the peak is not symmetric, which hints at there is more than more Pt species in the system. What are the possible coordinate environment of Pt on the support?

Second, the peak position is around 2070 cm^{-1} . This is in line with the works from some groups (e.g., *Chem* 2019, 5, 1–13; *Nat. Commun*, 2017, 8, 16100; *Nat. chem*, 2011, 3, 634–641, in which the peak is in the range of 2070–2090 cm^{-1}) but different from the others (*JACS* 2017, 139, 14150–14165; *Science*, 2015, 350, 189–192, in which the peak appear at >2100 cm^{-1}). This is most likely because of the charge state of Pt: the higher the valent state, the higher the wave numbers. Peak at around 2070 cm^{-1} suggest δ^+ in Pt1 is a rather small number. The authors may compare their data with literature ones and comment on the charge state of their Pt1 catalyst (or their very interesting metal-support interactions)

Response Letter

Reviewer #2 (Remarks to the Author):

1. The author used CO-DRIFT to demonstrate the single-atom identify of Pt. The spectra were provided in Figure S19. But a bit more discussion should be added to the text to enhance scientific level.

R: We would like to thank the referee for this comment. We added the following discussion about the results of CO-DRIFT to the revised main-text.

(Revised manuscript, page 10, line 17–20)

Here, the CO stretching frequency of 2070 cm^{-1} on 0.1Pt/C12A7 gave a slightly blue shift compared with other reported SACs. It can be ascribed to the basic surface of C12A7 with a strong electron donation effect, leading to a relative small δ^+ value in Pt.

2. First, the peak is not symmetric, which hints at there is more than more Pt species in the system. What are the possible coordinate environment of Pt on the support?

R: We would like to appreciate the referee for providing this nice comment. In our EXAFS results, derived Pt–O coordination number was estimated to be 3.34 and 2.25 in 0.1Pt/C12A7 and 0.1Pt/C12A7-R600 respectively, indicating the real coordination number of Pt–O should be 2 or 3, which can't rule out the possibility of the coexistence of two kinds of Pt coordination environments. Since the CO adsorption behavior is strongly related to the oxidation state of Pt, the higher the valent state of Pt resulted the higher wavenumber of CO adsorption peak in CO-DRIFT spectra. Therefore, we ascribed this symmetric of CO adsorption to the possible coexistence of Pt-O coordination numbers.

3. Second, the peak position is around 2070 cm⁻¹. This is in line with the works from some groups (e.g., Chem 2019, 5, 1–13; Nat. Commun, 2017, 8, 16100; Nat. chem, 2011, 3, 634-641, in which the peak is in the range of 2070-2090 cm⁻¹) but different from the others (JACS 2017, 139, 14150–14165; Science, 2015, 350, 189-192, in which the peak appear at >2100 cm⁻¹). This is most likely because of the charge state of Pt: the higher the valent state, the higher the wave numbers. Peak at around 2070 cm⁻¹ suggest delta+ in Pt1 is a rather small number. The authors may compare their data with literature ones and comment on the charge state of their Pt1 catalyst (or their very interesting metal-support interactions).

R: We appreciate the referee for attracting our attention to this issue. We highly agree with the reviewer's suggestion that the charge state of Pt determines the adsorption behavior of CO probe molecule. Indeed, previous reports of the vibrational frequency for linearly adsorbed CO to Pt single atom species vary with changes in the support such as Pt/TiO₂,¹ 2112 cm⁻¹; Pt/CeO₂,² 2095 cm⁻¹; Pt/m-Al₂O₃,³ 2084 cm⁻¹; and Pt/FeO_x,⁴ 2070 cm⁻¹). In our case, Pt/C12A7 exhibits the adsorption peak with the band at 2070 cm⁻¹, demonstrating a slightly blue shift compared with those reported results. In our previous work, C12A7 was reported to be a basic support and large amount of surface basic sites were confirmed through CO₂-TPD,⁵ resulting a strong electron donation effect on supported Pt. Therefore, these donated electrons may slightly reduce the charge state of Pt, casing a smaller δ+ value in Pt with a slight blue shifted CO stretching frequency.

References

1. DeRita, L. et al. Catalyst architecture for stable single atom dispersion enables site-specific spectroscopic and reactivity measurements of CO adsorbed to Pt atoms, oxidized Pt clusters, and metallic Pt clusters on TiO₂. *J. Am. Chem. Soc.* **139**, 14150–14165 (2017).
2. Ding, K. et al. Identification of active sites in CO oxidation and water-gas shift over supported Pt catalysts. *Science* **350**, 189–192 (2015).
3. Zhang, Z. et al. Thermally stable single atom Pt/m-Al₂O₃ for selective hydrogenation and CO oxidation. *Nat. Commun.* **8**, 16100 (2017).
4. Qiao, B. et al. Single-atom catalysis of CO oxidation using Pt₁/FeO_x. *Nat. Chem.* **3**, 634–641 (2011).

5. Ye, T. N., Li, J., Kitano, M. & Hosono, H. Unique nanocages of $12\text{CaO}\cdot 7\text{Al}_2\text{O}_3$ boost heterolytic hydrogen activation and selective hydrogenation of heteroarenes over ruthenium catalyst. *Green Chem.* **19**, 749–756 (2017).